# Natural immune boosting biases pertussis infection estimates in seroprevalence studies

Matthieu Domenech de Cellès ®[1,7] ✉, Anabelle Wong ®[1,2,7], Tine Dalby ®[3] & Pejman Rohani ®[4,5,6]

Seroepidemiology has significant potential for uncovering the unreported burden of infectious diseases. However, for diseases without well-defined serological correlates of protection, natural immune boosting—whereby pathogen exposure triggers a detectable immune response without causing a transmissible infection—can complicate the interpretation of serosurveys. This issue is relevant to pertussis, a vaccine-preventable disease that remains a significant public health concern worldwide. Here, we aimed to evaluate the reliability of pertussis serosurveys using a transmission model that tracked the dynamics of pertussis infection, natural immune boosting, and seroprevalence. By fitting this model to seroprevalence data from the late whole-cell pertussis vaccine era in six European countries, we estimated that protection against infection conferred by natural infection or vaccination was variable but lasted, on average, for several decades. We then predicted the positive predictive value (PPV) of seropositivity in serosurveys among adults across twelve countries that broadly captured transmission patterns worldwide. Overall, we predicted a low PPV across multiple scenarios, especially in adults aged 20–39 years, where it typically dropped below 50%. Thus, although serosurveys are unquestionably useful for quantifying pertussis exposure levels, the common interpretation of seroprevalence as a measure of recent infections may lead to overestimating pertussis circulation and underestimating the impact of pertussis vaccines.

Pertussis, also known as whooping cough, is a highly contagious respiratory disease caused predominantly by infection with the bacterium *Bordetella pertussis*, as well as other bacteria of the Bordetella genus[1]. Historically, this common childhood disease led to high infant mortality[2], until the development and widespread use of whole-cell pertussis (wP) vaccines significantly reduced reported cases during the second half of the twentieth century[3,4]. While wP vaccines remain recommended by the WHO and widely used globally, many high-income countries have switched to acellular pertussis (aP) vaccines

that became available in the 1990s[5–7]. Despite relatively high vaccine coverage worldwide (~ 85% for the primary series in the last ten years[8]), the burden of pertussis remains considerable, with an estimated 19.9 million new cases in children aged 0–14 in 2019[9]. Unexpectedly, a long-term resurgence of pertussis has been observed in several high-income countries with sustained high vaccination coverage, including the USA, Sweden, and Denmark[5]. Although reported pertussis cases plummeted shortly after the start of the COVID-19 pandemic[10–12], many countries—especially in Europe[13]—are now witnessing large epidemics, resulting in

[1]Max Planck Institute for Infection Biology, Infectious Disease Epidemiology group, Berlin, Germany. [2]Institute of Public Health, Charité—Universitätsmedizin Berlin, Berlin, Germany. [3]Department of Infectious Disease Epidemiology and Prevention, Statens Serum Institut, Copenhagen, Denmark. [4]Odum School of Ecology, University of Georgia, Athens, GA, USA. [5]Center of Ecology of Infectious Diseases, Athens, GA, USA. [6]Department of Infectious Diseases, College for Veterinary Medicine, University of Georgia, Athens, GA, USA. [7]These authors contributed equally: Matthieu Domenech de Cellès, Anabelle Wong. ✉e-mail: domenech@mpiib-berlin.mpg.de

infant deaths[14]. These alarming trends highlight the ongoing threats of pertussis, which remains one of the least controlled vaccine-preventable diseases worldwide.

A major challenge in pertussis epidemiological research is to estimate accurately the rates of pertussis infections. Standard surveillance systems often fall short because infections are reported only when patients exhibit symptoms, seek healthcare, and receive a clinical or laboratory diagnosis—with potential case loss at every step[15]. For pertussis, this problem is thought to be acute for at least three reasons. First, asymptomatic infections may be common, especially (but not only) among vaccinated age groups[16]. Second, clinical diagnosis based on typical pertussis symptoms—such as paroxysmal coughing, whooping, and posttussive vomiting—can be inaccurate[17]. Third, non-pediatricians may lack awareness of pertussis disease and fail to diagnose it in adult patients[18]. These factors collectively contribute to case underreporting, which is estimated to be substantial for pertussis[19–21].

Due to the limitations of standard surveillance data, sero-epidemiology is frequently used to assess the prevalence of antibodies against *B. pertussis* antigens, typically immunoglobulin G (IgG) against pertussis toxin (PT, a toxin unique to *B. pertussis*)[22]. However, since serological correlates of protection remain unidentified for pertussis[23], anti-PT IgG titers do not correlate well with immunity against infection. Indeed, anti-PT IgG antibodies from vaccination decrease to undetectable levels within a few years, while protection persists for longer[23]. This continued protection may be attributed to the relatively slow progression of pertussis infection (mean serial interval of ~3 weeks[24]), such that recall responses from memory cells may be rapid enough to provide partial protection, even in the absence of circulating antibodies[25]. Hence, in most seroprevalence studies, or serosurveys, seropositivity is interpreted as evidence of a recent infection, where the recency depends on the IgG threshold used to define seropositivity. As both symptomatic[26] and asymptomatic[27] infections generally induce an immune response, serosurveys can hypothetically quantify recent transmission levels, including asymptomatic infections. As a result, the baseline hypothesis of many serosurveys is that seroprevalence is a more accurate measure of pertussis circulation compared to case-based surveillance data, especially among adults[22]. Seemingly supporting this view, studies have found significant discrepancies between infection rates derived from serosurveys and those reported through surveillance data, with serosurveys often revealing infection rates that are much higher—sometimes by several orders of magnitude[22].

Several investigators, however, have questioned this view and cautioned that serological data may lack specificity[28,29]. This criticism is based on the fact that serology alone cannot distinguish between infection and natural immune boosting[30]. In other words, exposure to *B. pertussis* through contact with an infected host may trigger an immune boost—or anamnestic response—in individuals protected by earlier infection or vaccination, leading to seropositivity but not to a productive infection that can be transmitted to other hosts (*i.e.*, a transmissible infection). To clarify our terminology regarding the outcomes of *B. pertussis* exposures resulting in seropositivity, we henceforth restrict our definition of infection to an exposure leading to transmissible infection (either symptomatic or asymptomatic), and we define other exposures as natural immune boosts. We believe this definition is justified, as transmissible infections are arguably the most pertinent from an epidemiological and evolutionary perspective.

Empirical evidence supports the idea that immune boosting occurs for pertussis[31]. A household study conducted as part of the aP clinical trial in Sweden reported frequent observations of seropositivity without culture positivity[32]. Similarly, in a human challenge experiment in adults aged 18–45, the highest inoculum dose of 100,000 colony-forming units caused seroconversion in all participants; however, extensive environmental sampling could not detect any bacterial shedding, thus suggesting the absence of transmissible infection[27]. In a study of aP vaccines in adults over 50 years in Australia, vaccine effectiveness was substantially underestimated when cases were identified by single-titer serology, compared to PCR-confirmed cases; the authors interpreted this discrepancy as evidence of case misclassification and poor diagnostic specificity of serology in their setting, suggestive of immune boosting[33]. This body of evidence suggests that serosurveys may overestimate the incidence of pertussis infections.

Despite occasional mentions of these complexities[29,30], the potential unreliability of serosurveys and the implications of immune boosting on the interpretation of seropositivity are frequently overlooked in the literature (see our review of published serosurveys below). Due to this lack of awareness, the reliability of serosurveys has not been systematically examined. Resolving this knowledge gap is essential to reconcile the different estimates of pertussis infection from serosurveys and case-based surveillance and to evaluate the impact of pertussis vaccines more accurately. Here, we present results from a population-based model of pertussis transmission that tracked the dynamics of infection, immune boosting, and seropositivity and enabled a comparison with empirical seroprevalence estimates. We first calibrated the model based on seroprevalence data from the late wP vaccine era in six European countries. This calibration then allowed us, through a comprehensive simulation study, to predict the prevalence and reliability of seropositivity in various countries worldwide.

## Results

### Interpretation of seropositivity in published serosurveys

To understand how seroprevalence data were interpreted in the scientific literature, we first reviewed recent seroprevalence studies conducted on the general population (see Methods). We identified 59 articles published in the last five years and included 18 pertussis seroprevalence studies in our review. Through this analysis, we identified two broad categories of seroprevalence studies with distinct aims. In the first category of studies (Table S1), the investigators considered low to moderate anti-PT IgG thresholds to evaluate immune protection. However, the threshold to differentiate protection from lack of protection varied across studies, ranging from 5 IU/mL[34,35] to 50 IU/mL[36]. Furthermore, these studies were inherently limited because anti-PT IgG titers do not correlate well with protection, as the serological correlates of protection remain unidentified for pertussis[23]. Only two studies explicitly recognized this limitation[34,37].

In the second group of studies (Table S2), the investigators defined moderate to high anti-PT IgG thresholds to evaluate recent exposure or infection. With few exceptions[37–40], the threshold considered was 100 IU/mL; based on this cut-off, seropositivity was generally interpreted as evidence of exposure or infection within the last year. However, a few studies interpreted seropositivity as evidence of more immediate exposure or infection (acute infection[34,35] or within the past 58.6 days[41]), and many studies did not explicitly define what constitutes recency. In four studies[34,38,42,43], the term "exposure" was used instead of "infection," but no definition of exposure was provided. Notably, the issue of immune boosting and how it may result in false positives was not discussed in any study, highlighting a lack of awareness surrounding this problem.

### Estimation and model fit to seroprevalence data

To assess the reliability of pertussis seroprevalence studies, we developed a stochastic, simulation model of pertussis transmission that tracked the population-level dynamics of immune boosting and anti-PT IgG seroconversions, seropositivity, and seroreversions (see Fig. 1, Table 1, and Methods). We considered two model endpoints: the seroprevalence and the positive predictive value (PPV) of seropositivity, defined as the conditional probability of recent infection given seropositivity (or, equivalently, the proportion of true cases among seropositive cases in serosurveys).

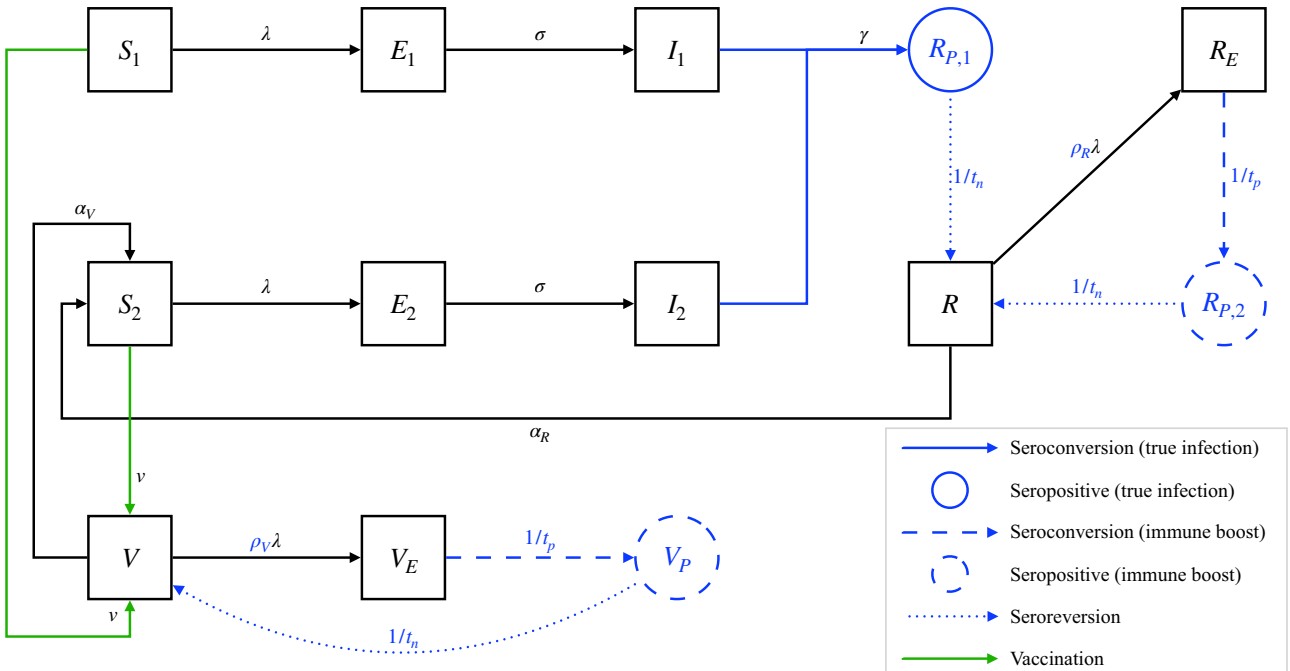

**Fig. 1 | Schematic representation of the serotransmission model.** For clarity, only one age group is depicted, and demographic transitions such as birth, aging, and death are not represented. The serological parameters are highlighted in blue, and the seropositive states are circled in blue (see Table S3 for the definition of all state variables). Following either primary (state variables $S_1, E_1, I_1$) or secondary infection (state variables $S_2, E_2, I_2$), individuals are assumed to remain seropositive for an average duration $t_n$ (state variable $R_{P,1}$). Upon exposure to *B. pertussis*, seronegative individuals with either infection-derived (state variable $R$) or vaccine-derived immunity (state variable $V$) against infection may undergo an immune boost and become seropositive (state variables $R_P$ and $V_P$, respectively). State variables with an E index represent individuals exposed to *B. pertussis* and about to seroconvert. The diagram highlights a key issue with seroprevalence studies: generally, seroprevalence (represented as $S_p = R_{P,1} + R_{P,2} + V_P$, the sum of the blue states) includes both recent infections ($R_{P,1}$) and immune boosts from individuals with immunity derived from earlier infection ($R_{P,2}$) or vaccination ($V_P$). Consequently, seropositivity does not always indicate a recent infection, which may lead to false positives and a low positive predictive value (defined here as $PPV = R_{P,1}/S_p$). In this model, infections are defined by their ability to transmit to other hosts, whereas immune boosts do not contribute to transmission.

## Table 1 | Main model parameters

| Parameter | Meaning | Value(s) | Source/Comment |
|---|---|---|---|
| $\sigma^{-1}$ | Average latent period | 8 days | 64 |
| $\gamma^{-1}$ | Average infectious period | 15 days | 64 |
| $\theta$ | Relative infectiousness of secondary infections | 0.99 | 64 |
| $N_i$ | Age-specific population sizes | Fig. S4 | 46<br>Country-specific |
| $M_{ij}$ | Social contact matrix | Fig. S3 | 46<br>Country-specific |
| $R_0$ | Basic reproduction number | 11–16 | Country-specific |
| $t_p$ | Average time from exposure to seroconversion | 23 days | 26 |
| $t_n$ | Average duration of seropositivity after seroconversion | 1 yr (seropositivity cut-off of 100 IU/mL)<br>0.75 yr (seropositivity cut-off of 125 IU/mL) | 26 |
| $P_V$ | Vaccination coverage (all doses) | 0.95 | Assumption |
| $\epsilon_V$ | Probability of primary vaccine failure | 0.05 | 64 |
| $p_V = P_V(1-\epsilon_V)$ | Effective vaccination coverage (all doses) | 0.9 | Table S6 |
| $v$ | Effective vaccination rate | Calculated from $p_V$ | Supplementary methods |
| $\rho$ | Boosting coefficient of infection/wP-derived immunity | 0.5, 1, 2, 5 | Range of values tested based on earlier studies[31,50–52]<br>Assumption: $\rho = \rho_V = \rho_R$ |
| $\alpha^{-1}$ | Average duration of infection/wP-derived immunity (if no immune boosting) | Estimated based on empirical serosurveys in the late wP era | Assumption: $\alpha^{-1} = \alpha_V^{-1} = \alpha_R^{-1}$ |

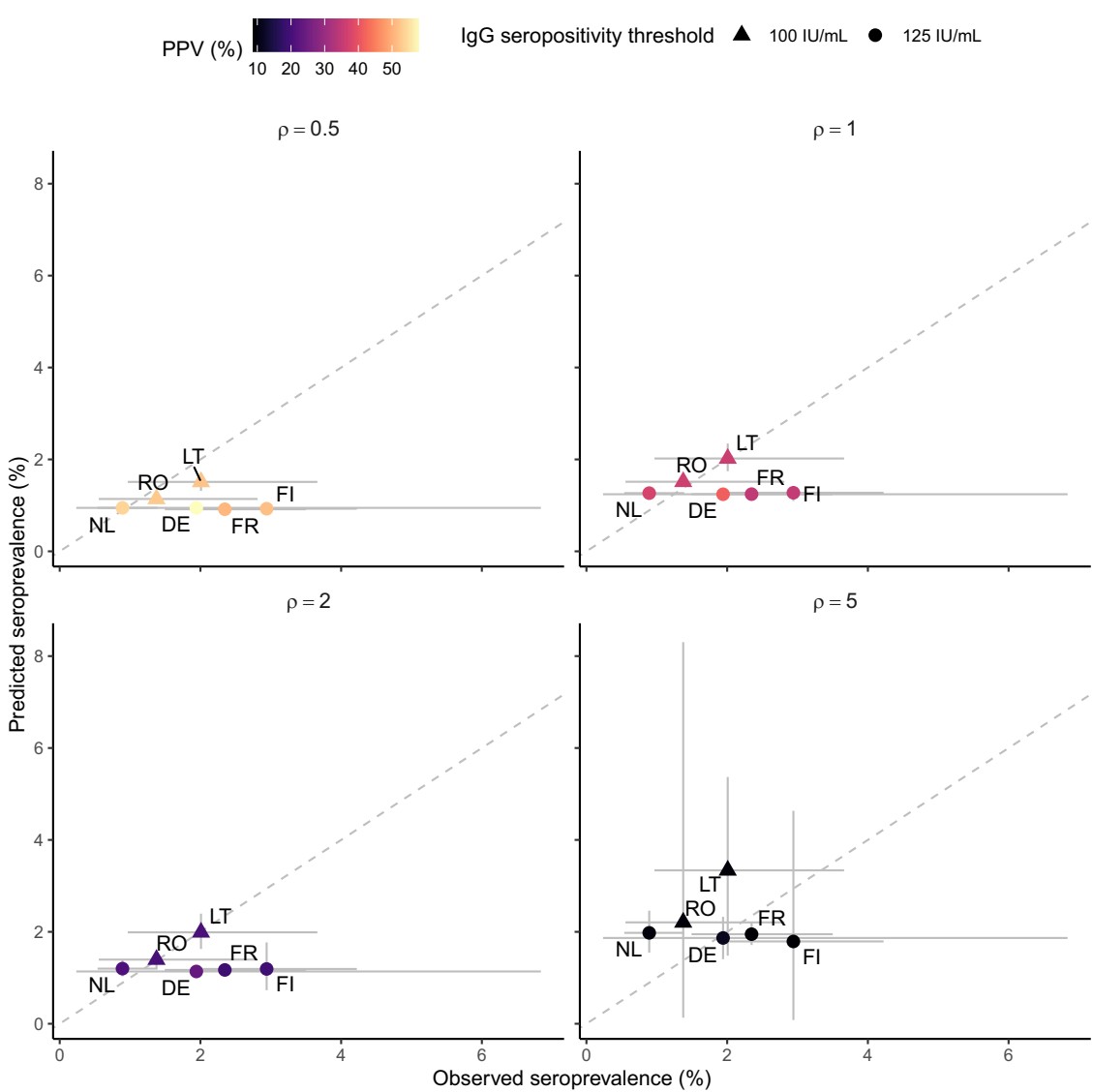

**Fig. 2 | Model fit to empirical seroprevalence data in six European countries.**
PPV positive predictive value of serology, IU international unit. The dashed line is
the identity line, representing a perfect match between the model and the data. The
2-letter codes indicate the countries considered for model-data comparison: DE
(East Germany), FI (Finland), FR (France), LT (Lithuania), NL (Netherlands), and RO
(Romania). The data and predictions represent seroprevalence in adults aged
20–39 in Lithuania and Romania[45] or 20–44 in other countries[44]. For each immune-
boosting level ($\rho$), the graph shows the model predictions for the duration of
infection/wP-derived immunity that produced the best fit to the seroprevalence
data (see Table 2 for the corresponding value). The intervals represent 95% con-
fidence intervals for the data (x-axis) and 95% prediction intervals for the model
simulations (y-axis).

We first aimed to parametrize this model by fitting it to ser-
oprevalence data from two large serosurveys conducted during the
late wP era in Europe[44,45] (Methods). After applying our inclusion cri-
teria (Table S5), we selected six countries for the model-data com-
parison of seroprevalence. These included four countries (Finland,
France, East Germany, and the Netherlands) from ref. 44 and two
countries (Lithuania and Romania) from ref. 45. Serological samples
were collected approximately 30–45 years after the start of wP vacci-
nation in the first four countries and 50 years after in the latter two. In
all six countries, the primary vaccine series started 2–3 months after
birth with an additional booster dose at age 1–3 years, and the vacci-
nation coverage remained high (over 85%) until the serosurvey
(Table S6). Keeping in mind the different seropositivity thresholds
used in the two studies (100 IU/mL and 125 IU/mL), the seroprevalence
estimates were fairly consistent across countries, ranging from 0.9%
(standard error [SE]: 0.2 %) in the Netherlands to 2.9% (SE: 0.5%) in
Finland.

The model fit to seroprevalence data for every immune-boosting
scenario is displayed in Fig. 2. The average duration of infection/wP-
derived protection was estimated at several decades in all scenarios,
with point estimates ranging from 30 to 50 years (Table 2). However,
the duration of protection was inherently variable, such that these
averages translated into a sizeable fraction of 10–15% of vaccinees
losing protection within 5 years and 18–28% within 10 years. These
estimates resulted in a low mean weighted error (MWE) of < 0.5% in
every scenario, demonstrating strong model-data agreement irre-
spective of the fixed boosting level. Hence, the immune-boosting
coefficient could not be estimated from these data. Across the sce-
narios, the PPV of serology decreased as immune boosting increased,
with a range (across countries) of 47–62% for the lowest boosting level
and 6–12% for the highest (Table 2). These results suggest that the
seroprevalence data were consistent with a long, although variable
duration of protection, with immune boosting accounting for a large
part of seropositive cases.

**Table 2 | Estimates of the average duration of immunity**

| Fixed value of boosting coefficient ($\rho$) | Estimate (95% confidence interval) of the average duration of infection/wP-derived immunity ($\alpha^{-1}$), years | MWE, % | PPV range, % |
|---|---|---|---|
| 0.5 | 30 (20–40) | −0.4 | 47–62 |
| 1.0 | 30 (30–50) | 0.0 | 29–41 |
| 2.0 | 40 (30–60) | −0.1 | 16–27 |
| 5.0 | 50 (40–80) | 0.1 | 6–12 |

MWE stands for mean weighted squared error, while PPV represents the positive predictive value of serology. The uncertainty interval indicates the range of values for which the MWE was not significantly different from zero. The PPV range indicates the range of median PPV across the six countries included in the model-data comparison and across scenarios where the MWE was non-significant.

## NGM clustering and representative countries

To get a broader picture of the potential shortcomings of serology, we then used our model to predict the seroprevalence and PPV of seropositivity across various countries worldwide. To simplify our analysis, we clustered the next-generation matrices (NGMs) derived from Mistry et al.[46] to identify groups of countries with broadly similar transmission dynamics.

The clustering analysis identified 11 clusters among the 35 country-level NGMs. Because of the large number of serosurveys conducted in China, we separated the China NGM (initially grouped with the NGMs from Australia, Canada, and the US), resulting in 12 clusters overall. As shown in the resulting dendrogram (Fig. S2), the number of countries in each cluster varied, ranging from 1 (Israel, China) to 5. Some clusters comprised countries from the same geographical region (e.g., the Norway-Denmark and France-Italy-UK clusters), while others included geographically distant countries (e.g., the India-South Africa cluster). To reduce our subsequent analysis, we selected one representative country from each cluster, resulting in the following 12 countries: China, Czechia (aka the Czech Republic), Denmark, Germany, India, Israel, Japan, the Netherlands, Sweden, Switzerland, the UK, and the USA. The corresponding social contact matrices (SCMs) are plotted in Fig. S3.

## Variations in seroprevalence and PPV across 12 representative countries

To assess the prevalence and reliability of seropositivity more broadly, we simulated our calibrated model in the 12 representative countries identified by NGM clustering. The model reached equilibrium across all boosting levels in the 12 countries (see Fig. S5 for representative time series in the USA). Even though our model did not include seasonality in transmission, the simulated seroprevalence exhibited multiannual cycles in the hyper-sensitive boosting scenarios ($\rho > 1$). This behavior has been reported in previous theoretical analyses of hyper-sensitive boosting models[31,47].

Figure 3 shows the variation in seroprevalence and PPV across countries, age groups, and immune-boosting levels based on an average duration of protection of 40 years (consistent with the seroprevalence data for all boosting levels; see Table 2). The predicted seroprevalence ranged from 0.5 to 2.5%, displaying large variability across scenarios. The key factors contributing to this variability were age and immune-boosting levels, with seroprevalence predicted to increase as age decreased (by a factor 2–4 for 20–39 yo in comparison to 60–79 yo) or immune boosting intensified (by a factor 1.5–3.0 for the highest boosting level compared with the lowest). In contrast, variations in SCMs resulted in less variability in seroprevalence.

The PPV of serology in seroprevalence studies ranged from 10 to 80% across scenarios. As for the seroprevalence, the PPV varied sensitively with the strength of immune boosting and age. Specifically, the PPV increased with age, by a factor 1.8–3.8 for 60–79 yo in comparison to 20–39 yo; in contrast, the PPV decreased as the immune-boosting strength increased, by 40–80% when comparing the highest boosting level to the lowest. Consequently, seropositivity was predicted to be most reliable in the elderly (60–79 yo) when boosting was low and least reliable in young adults (20–39 yo) when boosting was high.

To further characterize the reliability of serosurveys as a function of age and immune-boosting strength, we ran additional simulations of seroprevalence and PPV across all scenarios consistent with the empirical seroprevalence data (Table 3). Overall, the median predicted seroprevalence from the model was in the range 0.6–2.2% in 20–39 yo, 0.4–1.8% in 40–59 yo, and 0.2–1.1% in 60–79 yo. These additional simulations confirmed the variations observed in Fig. 3: for fixed duration of protection and immune-boosting strength, as age increased, seroprevalence decreased while the PPV of serology increased; in contrast, when the duration of protection and age were held constant, stronger boosting resulted in higher seroprevalence but lower PPV. Additionally, at fixed age and immune-boosting level, a shorter duration of protection led to both higher seroprevalence and PPV of serology. These results illustrate that the reliability of serology in seroprevalence studies varies with age and the characteristics of immunity, which complicates the interpretation of serosurveys when these characteristics are unknown.

To better understand how the prevalence and reliability of seropositivity vary with age, we dissected seroprevalence across different age groups (Fig. 4). At a fixed boosting level (e.g., $\rho = 1$), the force of infection decreased with age (from 0.6% per yr in 20–39 yo to 0.3% per yr in 60–79 yo, Fig. 4A), reflecting general age-specific contact patterns, particularly the lower contact rates in the elderly (Fig. S3). Meanwhile, the fraction susceptible to infection increased with age (from ~35% in 20–39 yo to ~70% in 60–79 yo) due to the gradual loss of infection-/wP-derived protection (Fig. 4C). As a result of these two effects, the seroprevalence due to true infections peaked in the intermediate age group of 40–59 yo (Fig. 4D). Conversely, the seroprevalence resulting from immune boosts declined as age increased (from 0.6% in 20–39 yo to 0.1% in 60–79 yo, Fig. 4F), as loss of immunity gradually reduced protection against infection and thus opportunities for immune boosting in older age groups (Fig. 4E). Overall, this decline outweighed the age variations in seroprevalence caused by true infections so that the overall seroprevalence decreased with age (Fig. 4B). Consequently, seropositivity was less frequent but also more likely to represent a true infection because of reduced immune boosts in older age groups. These results highlight the complexity of interpreting seropositivity and suggest that seroprevalence may not even qualitatively capture age variations in infection.

We repeated these simulations to examine how immune-boosting strength affects seroprevalence (Fig. 4). In a given age group (e.g., 20–39 yo), the force of infection was predicted to decrease as the strength of immune boosting increased (range from lowest to highest boosting level: 0.7% per yr to 0.2% per yr, Fig. 4A). This effect can be explained as follows: assuming all other factors remain constant, increasing the strength of immune boosting is analogous to extending the duration of protection (Fig. 1), resulting in reduced circulation and a lower prevalence of infection—and thus a lower force of infection—across age groups. As a result, the seroprevalence due to true infections decreased with stronger immune boosting (range: 0.3% to 0.2%,

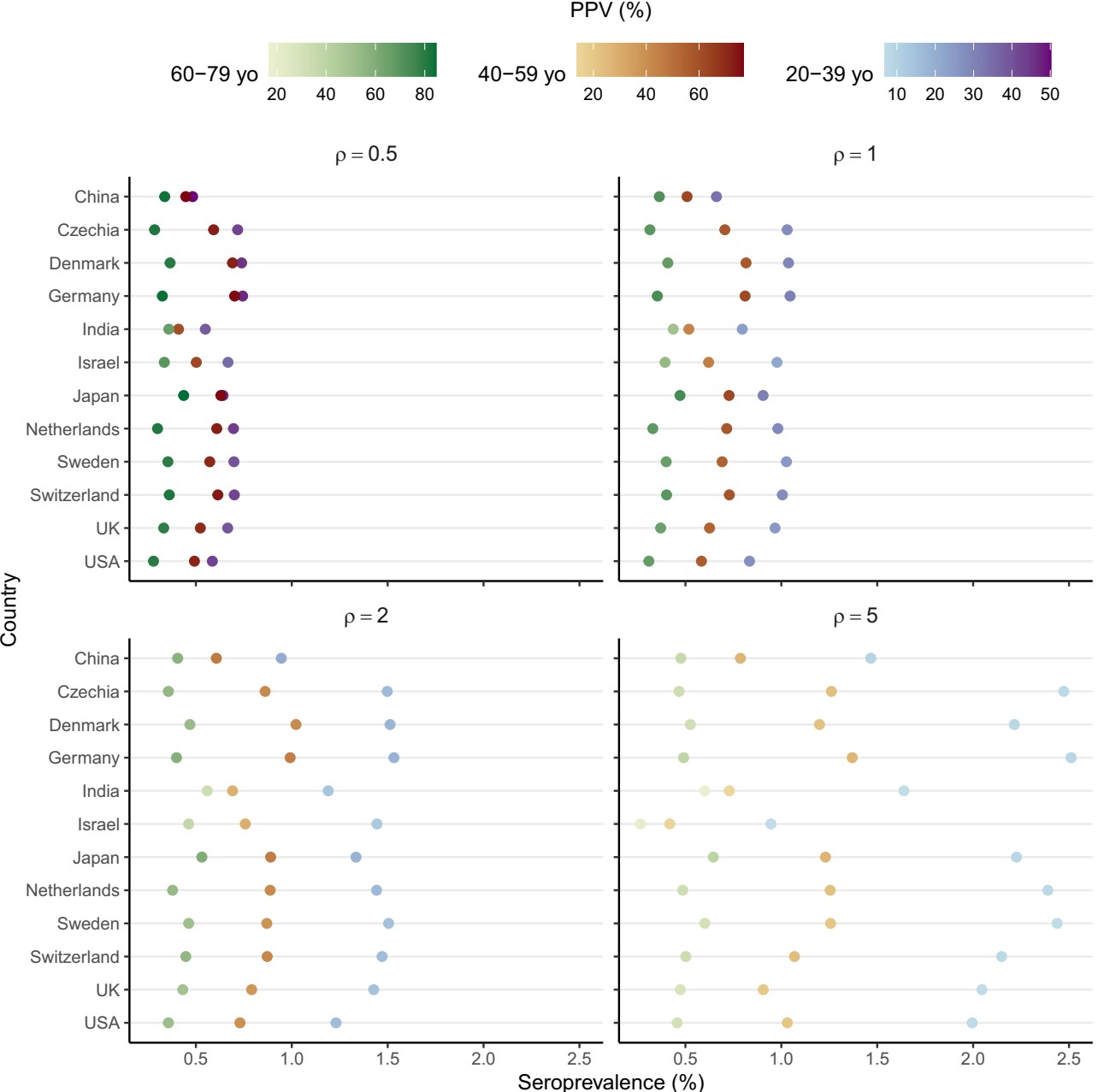

**Fig. 3 | Predicted seroprevalence in 12 representative countries.** In each country, the location of the points indicates the median seroprevalence from 200 simulation-years (10 model simulations, each spanning 20 years). The panel title specifies the fixed value of the boosting coefficient for infection- or vaccine-derived immunity ($\rho$), while the color scales represent the positive predictive value (PPV) in three different adult age groups (20–39 yo, 40–59 yo, and 60–79 yo). The average duration of infection- or wP-derived immunity was set to $\alpha^{-1} = 40$ years in all simulations, a value within the confidence interval for all tested values of the boosting coefficient (see Table 2).

Fig. 4D). In contrast, the seroprevalence from immune boosts increased with immune-boosting strength (range: 0.3% to 1.8%, Fig. 4F). This increase dominated the changes in seroprevalence due to true infections, so that the overall seroprevalence increased with immune-boosting strength (range: 0.6% to 2.0%, Fig. 4B). Thus, higher levels of immune boosting resulted in fewer true positives (i.e., seropositive cases after a true infection) and more false positives (i.e., seropositive cases after an immune boost). By definition of the PPV, these two effects combined to reduce the PPV of serology in seroprevalence studies (as also observed in Fig. 3 and Table 3).

In conclusion, these results illustrate how the complex interplay between waning protection, immune boosting, and the age-specific contact patterns and force of infection sensitively determines seroprevalence and the reliability of seropositivity as an indicator of recent infection in seroprevalence studies.

### Alternative model structure and parametrization of immune boosting

To test the robustness of our results, we considered an alternative model structure in which immune boosting was allowed to decline more gradually with time since infection or vaccination (see model schematic in Fig. S6). In this model, the duration of vaccine-derived protection (without boosting) followed a Gamma distribution (with shape parameter 2 and mean $1/\alpha$), while the duration of infection-

**Table 3 | Predicted seroprevalence and positive predictive value of serology across 12 representative countries**

| Boosting coefficient ($\rho$) | Average duration of immunity ($\alpha^{-1}$) | Seroprevalence (%) | | | Positive Predictive Value (%) | | |
|---|---|---|---|---|---|---|---|
| | | 20–39 yo | 40–59 yo | 60–79 yo | 20–39 yo | 40–59 yo | 60–79 yo |
| 0.5 | 20 yr | 2.0 (1.9–2.3) | 1.8 (1.6–2.2) | 1.1 (0.9–1.4) | 52 (43–58) | 77 (70–79) | 83 (75–84) |
| | 30 yr | 1.1 (0.9–1.3) | 1.0 (0.8–1.2) | 0.6 (0.5–0.7) | 47 (38–53) | 75 (64–78) | 82 (70–84) |
| | 40 yr | 0.7 (0.5–0.8) | 0.6 (0.4–0.8) | 0.3 (0.3–0.4) | 43 (35–50) | 72 (61–77) | 81 (66–85) |
| 1.0 | 30 yr | 1.5 (1.3–1.8) | 1.1 (0.9–1.4) | 0.6 (0.5–0.8) | 31 (24–37) | 61 (48–65) | 70 (54–74) |
| | 40 yr | 1.0 (0.7–1.2) | 0.7 (0.5–0.9) | 0.4 (0.3–0.5) | 28 (21–34) | 57 (44–63) | 68 (50–74) |
| | 50 yr | 0.6 (0.3–0.8) | 0.4 (0.2–0.6) | 0.2 (0.1–0.3) | 26 (20–32) | 54 (41–60) | 66 (46–73) |
| 2.0 | 30 yr | 2.2 (1.7–2.7) | 1.4 (1.0–1.8) | 0.7 (0.5–1.1) | 19 (14–23) | 45 (32–50) | 56 (38–61) |
| | 40 yr | 1.4 (0.9–1.9) | 0.8 (0.5–1.3) | 0.4 (0.3–0.7) | 17 (12–21) | 41 (28–47) | 53 (34–60) |
| | 50 yr | 0.9 (0.4–1.4) | 0.5 (0.2–0.9) | 0.3 (0.1–0.4) | 15 (11–19) | 38 (26–44) | 50 (30–58) |
| 5.0 | 40 yr | 2.2 (0.1–5.3) | 1.2 (0.0–3.5) | 0.5 (0.0–2.3) | 8 (6–10) | 23 (13–27) | 33 (16–40) |
| | 50 yr | 1.4 (0.0–3.7) | 0.7 (0–2.3) | 0.3 (0.0–1.1) | 7 (5–10) | 20 (12–24) | 29 (14–38) |
| | 60 yr | 0.8 (0.0–2.9) | 0.4 (0.0–1.8) | 0.2 (0–0.8) | 6 (3–9) | 18 (5–22) | 26 (7–35) |
| Overall range | | 0.0–5.3 | 0.0–3.5 | 0.0–2.3 | 3–58 | 5–79 | 7–84 |

The values indicate the median (95% prediction intervals) from 200 simulation-years (10 model simulations, each spanning 20 years) across the 12 countries.

derived protection followed a generalized Erlang, or hypoexponential, distribution (*i.e.*, the sum of three exponential distributions with rates $1/t_n$, $2\alpha$, and $2\alpha$). In both cases, therefore, the distribution of the duration of protection was assumed less variable than that of the base model. For simplicity, we ignored age structure (homogeneous model); we considered two target levels of overall seroprevalence, based on two nationwide serosurveys (both with an anti-PT IgG seropositivity threshold of 62.5 IU/mL): 10% (Netherlands, 2006–2007[48]) and 20% (Australia, 1997–1998[49]). To our knowledge, these estimates are the highest reported in the vaccine era. As shown in Table S7, our main results remained robust, even for the lowest levels of immune boosting and the highest seroprevalence: a combination of relatively long average duration of protection (range: 30–60 years) and low PPV (range: 8–25%) best explained the empirical estimate. Hence, these additional simulations suggest that our main results are robust to different modeling assumptions regarding waning protection and boosting.

## Discussion

In this study, we aimed to evaluate the reliability of pertussis seroprevalence studies, particularly how the well-documented phenomenon of natural immune boosting may lead such studies to overestimate pertussis infections. To address this, we developed a new model of pertussis transmission that tracked the dynamics of seroprevalence, enabling a comparison with empirical estimates from serosurveys. Fitting this model to two large European serosurveys in the late wP era, we estimated infection-/wP-derived protection to last, on average, for several decades (keeping in mind that this duration was variable and much shorter in a sizeable fraction of the population). We then predicted the prevalence and PPV of seropositivity among adult age groups in twelve countries that were broadly representative of transmission patterns worldwide. Overall, we predicted a low PPV of serology for seroprevalence studies across multiple scenarios, especially in young adults aged 20–39, where it fell below 50% in almost all scenarios tested. We conclude that the issue of immune boosting is likely severe, making raw seroprevalence estimates potentially misleading when interpreted in isolation. Our model can be useful for analyzing seroprevalence data, ideally in conjunction with disease incidence data to synthesize all available evidence and derive more accurate estimates of pertussis infections.

When comparing our model to seroprevalence data from two large European serosurveys, we found the best model-data agreement for average durations of infection-/wP-derived protection of 30–50

years (overall range of 95% confidence intervals: 20–80 years). As our model did not include several real-world complexities of pertussis (such as seasonality in transmission, variations in vaccine coverage or changes in demographic structure), we emphasize this estimation is only approximate. Nevertheless, these durations are consistent with previous estimates from immune-boosting models fitted to case report incidence data. Wearing & Rohani estimated that durations of infection-/wP-derived protection of 20–40 yr best reproduced the interepidemic periods, and of 40–100 yr the patterns of epidemic fade-outs observed in the wP era in England and Wales (assuming an immune boosting level of 0.5 and an exponential distribution for the duration of protection)[50]. Based on incidence data in the prevaccine era in Copenhagen, Denmark, Lavine et al. estimated infection-derived protection to last on average 34 (95% CI: 17–66) years (assuming a Gamma distribution for the duration of protection), a value well identified despite minimal information regarding the immune-boosting parameter (6.6, 95% CI: 0.66–66)[51]. Using the same model, Rozhnova et al. found that an average duration infection-derived protection of 40 years (assuming an immune boosting coefficient of 1) resulted in periodic epidemic patterns consistent with those observed in the prevaccine era in Ontario, Canada, and London, UK[52].

Although differences in model structure preclude an exact comparison of all available estimates, our results add to the large body of modeling and epidemiological evidence about the long (but variable) protection conferred by natural infection and wP vaccines against pertussis infection and the marked impact of wP vaccines on pertussis transmission[53–55]. They also suggest that, due to natural immune boosting and the high transmissibility of pertussis, seroprevalence estimates of a few percent in adults are expected for imperfect yet highly effective vaccines, even in populations with near-perfect pediatric vaccination coverage.

In contrast to the duration of protection, all tested values for immune boosting resulted in an equally good model-data agreement. Hence, we could not estimate this parameter based on seroprevalence data alone. Other attempts to identify this parameter based on disease incidence data yielded similarly uncertain estimates[31,51,56]. A lower bound of 0.66 was identified in Lavine et al.'s study in Copenhagen[51]. A lower bound of 10 was estimated based on prevaccine era data in Massachusetts, USA[31], but a subsequent study in the USA reported such levels to be too high to reproduce the observed patterns of pertussis resurgence from the 1970s[57]. Given this admittedly limited evidence, we believe the range we considered (0.5–5) is reasonable, but we acknowledge the remaining uncertainties. Consequently, a

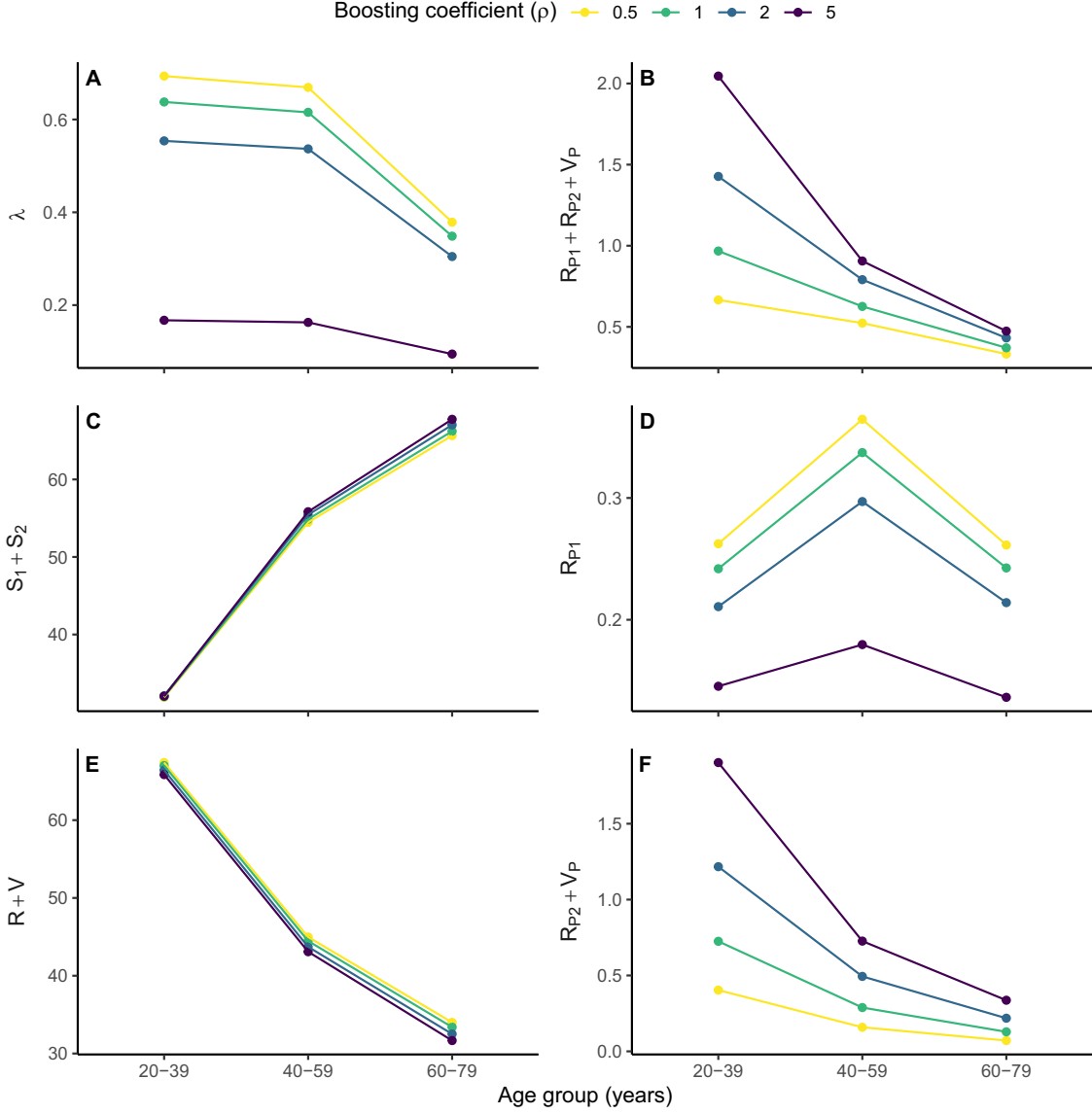

**Fig. 4 | Breakdown of seroprevalence and illustration of age-specific transmission dynamics in the US.** The values shown represent the median across 200 simulation years (derived from 10 model simulations, each spanning 20 years) for six model variables indicated by the *y*-axis titles (see Fig. 1): **A** force of infection (% per year); **B** overall seroprevalence (%); **C** fraction susceptible to infection (%); **D** seroprevalence due to true infections (%); **E** fraction immune to infection (%); **F** seroprevalence due to immune boosts (%). The color indicates the fixed value of the boosting coefficient for infection- or wP-derived immunity ($\rho$). In all simulations, the average duration of infection- or vaccine-derived immunity was set to $\alpha^{-1} = 40$ years, a value within the confidence interval for all tested values of the boosting coefficient (see Table 2). The *y*-axis values differ between panels.

promising avenue for future research will be to fit our model to multiple real-world data sources to more accurately estimate the level of immune boosting and the rates of pertussis symptomatic and asymptomatic infections.

Our results suggest that the interpretation of pertussis seroprevalence estimates is challenging and must be placed within a broader epidemiological context. Specifically, we find that the prevalence and reliability of seropositivity result from a complex interplay between immune boosting, waning protection, and age-specific contact patterns. In particular, we predict that the PPV of serology in seroprevalence studies varies predictably with age, with the lowest values in young adults (20–39 yo) who more often experience immune boosting. Strikingly, in almost all scenarios tested, the PPV fell below 50% in this age group. In other words, interpreting seropositivity as evidence of a recent infection may be incorrect more than half of the time in this age group. This finding is noteworthy, as our review

showed that this interpretation was nearly universal in seroprevalence studies. In the older age groups (40–59 yo, 60–79 yo), the reliability of seropositivity was better but still low in many scenarios, with the maximum PPV ranging from 79–85% at the lowest boosting level to 27–40% at the highest. Hence, in line with earlier suspicions[28–30] and empirical evidence[27,32,33], our results highlight that the issue of immune boosting must be considered when analyzing pertussis seroprevalence data. More broadly, our study emphasizes the key difference between exposure and infection, as seropositivity always indicates the former but not necessarily the latter. Therefore, our results do not imply that serosurveys are irrelevant, as seroprevalence data can accurately quantify the level of pertussis exposure in the population.

Another implication of our study is that comparing case report data to seroprevalence data will *not* yield valid estimates of underreporting. As immune boosting will generally cause seroprevalence to overestimate true infection rates, this ratio will tend to underestimate

the reporting probability or, equivalently, overestimate under-reporting, in turn leading to an overly pessimistic assessment of the impact of pertussis vaccines on pertussis circulation.

Our study has several important limitations that relate to the formulation of our model. First, we assumed that the serological parameters did not vary with age, as the data available[26] did not permit an age-specific parametrization. However, age variations in the immune response—*e.g.*, because of immunosenescence—may cause these parameters to differ between age groups, especially in the elderly[58]. Second, given the limited information in seroprevalence data, we made the simplifying assumption that infection- and wP-derived immunities had identical properties (in terms of immune boosting and duration of protection). This assumption is justified by epidemiological evidence[56,57] and immunological evidence showing that natural infection and wP vaccination trigger a comparable immune response[59,60]. Nevertheless, this assumption would be invalid for aP vaccines, which trigger a different immune response[61] that may be less prone to subsequent boosting by natural exposure[62]. Epidemiological evidence also shows that these vaccines, though effective at reducing transmission and inducing herd immunity[5,63], confer shorter-lived protection against infection, with waning rates of 2–10% per year[57,64–66]. Hence, extending our model to the aP era would require a separate parametrization for infection/wP-derived and aP-derived immunities. This complication prevented us from considering more recent seroprevalence data, especially those from another large serosurvey by Berbers et al. in adults aged 40–59 conducted in 18 European countries in 2015[42]. Still, we note that the seroprevalence estimates in Berber's study (range: 2.7–5.8% across 13/18 countries) were only moderately larger than those considered in our study.

Other limitations of this study relate to our estimation method. First, we only considered data from two large serosurveys conducted in European countries, as their serological results were standardized and ensured comparability across countries. However, many other seroprevalence data are available and could be used to estimate the parameters and test the predictions of our model[22]. Nevertheless, seroprevalences of similar magnitude (a few percent) have been estimated among adults in various other countries (*e.g.*, in Australia[67], Israel[68], Japan[22], and the USA[22]), so we believe our results should remain robust even with the inclusion of additional data. Second, as discussed above, when fitting our model to seroprevalence data, we did not consider seasonal or long-term parameter variations that may affect seroprevalence. However, in all countries included in our analysis, sample collection took place over ~1 year, allowing for any seasonal effects, if present, to be smoothed out. In addition, we did not include other data to inform the model about the past epidemics and age-specific history of prior exposures and infections in each country. In practice, the serosurveys may have captured a different stage of the epidemic cycle in every country, yet our model predictions were based on averages across multiple epidemic cycles (Fig. S5). Thus, our estimates are approximate, though they generally agree with previous estimates from more detailed models. Third, we assumed identical properties of wP vaccines across the six countries, although these vaccines were produced by different national or commercial manufacturers during the study period and might have had different efficacies[44,69]. Finally, even though we tested a range of realistic values, we could not estimate the strength of immune boosting, a parameter that sensitively controls the prevalence and reliability of seropositivity. As noted above, additional information in the form of case-based data should help resolve this uncertainty. Acknowledging these limitations, our model could serve as a building block to investigate the remaining unknowns in pertussis epidemiology.

There is growing interest in collecting serological data to inform the immunity landscape and tap into the "epidemiological dark matter" of infectious diseases[70–72]. For pathogens with identified correlates of protection (like neutralizing antibodies for measles and mumps[73,74]),

serology is relatively unequivocal and can directly inform specific variables of transmission models, such as recovered or vaccinated compartments. In the case of pertussis and other pathogens[70], however, the picture is much more complex, as seropositivity indicates a recent exposure. As a result, interpreting serological data is inherently ambiguous and requires careful consideration of waning immunity and immune boosting using transmission models. Hence, our model—or variations thereof—may prove useful for analyzing seroprevalence data and synthesizing evidence from other sources, including disease notification data. Eventually, fitting such models to all available data in multiple populations will improve our estimates of pertussis infection rates and help resolve the ongoing disagreements within the field.

## Methods

### Review of the interpretation of seropositivity in published serosurveys
We aimed to review recent seroprevalence studies conducted on the general population to understand how seroprevalence data were interpreted in the scientific literature. We specifically examined how investigators interpreted seropositivity and whether they addressed the issue of natural immune boosting.

Following the search terms used by a previous review by Barkoff et al.[22], we conducted a search for articles published in the past five years (as of the search date, February 3, 2025) that contain [pertussis AND seroprevalence], [pertussis AND serosurvey], [pertussis AND serosurveillance] OR [pertussis AND seroincidence] in the Title/Abstract on the literature database, PubMed.

We initially excluded records based on article type and language. In the abstract retrieval and screening stage, we further excluded records due to their (i) study type (assay method development); (ii) study aim (investigating antibody waning among vaccinated children or examining factors associated with antibody concentrations instead of assessing seroprevalence in the sample); or (iii) being conducted in specific risk groups (pregnant women, healthcare workers [HCW], or patients with chronic obstructive pulmonary disease [COPD]). Full-text articles of the remaining records were retrieved and assessed for eligibility, resulting in additional studies being excluded due to their (i) article type, (ii) study type (simulations or modeling), (iii) study aim (antibody waning among vaccinated children or factors associated with antibody concentrations), or (iv) being conducted in specific risk groups (adults of child-bearing age). Study screening was conducted by one reviewer. Data were extracted from the included articles by two reviewers independently and the extraction was cross-checked by both reviewers. The studies' identification process is summarized in the PRISMA flow diagram displayed in Fig. S1 (see also the PRISMA-P 2020 reporting checklist at the end of the supplement).

### Model structure
We developed a stochastic simulation model of pertussis transmission that generated serological endpoints in addition to standard prevalence and incidence endpoints (Fig. 1). The backbone of this model was similar to that of a previous model developed to explain the epidemiology of pertussis in the USA[64,65]. Briefly, the model distinguished between primary infections in fully susceptible individuals (*i.e.*, never vaccinated or infected) and secondary infections in susceptible individuals whose immunity was primed by earlier vaccination or infection. Based on earlier evidence in the USA[64,65], we assumed that vaccine-derived protection against infection was imperfect and might immediately fail (primary vaccine failure), with subsequent waning of protection. Similarly, infection-derived protection against infection was assumed to wane over time.

To generate serological endpoints, the model tracked the population-level dynamics of anti-PT IgG seroconversions, seropositivity, and seroreversions (Fig. 1). Unless otherwise stated, seropositivity was defined as an antibody titer exceeding 100 IU/mL, a

standard threshold used in numerous seroprevalence studies (refs. 22,75 and Table S2). Following recovery from either a primary or secondary infection, seropositivity was assumed to occur and last, on average, $t_n$ years. In individuals with either infection- or vaccine-derived protection, exposure to *B. pertussis* also led to seropositivity (but not to transmissible infection) at a rate proportional to the force of infection $\lambda$. Because such seroconversions arise due to natural immune boosting, the proportionality constant $\rho$ is called the *immune-boosting coefficient*[31]. This parameter controls the sensitivity of immune boosting: for $\rho < 1$, an exposure sufficient for infection is less likely to induce immune boosting, while for $\rho > 1$ an exposure insufficient for infection can still result in immune boosting. The latter scenario, known as hypersensitive boosting, can also be interpreted as follows: an exposure dose of *B. pertussis* antigen lower than what is needed for infection can still result in immune boosting. For completeness, we considered the former scenario of hyposensitive boosting, but we note this assumption is conservative and immunologically unlikely.

We note that, in this model, seropositivity implies protection from infection; the converse, however, is not true, as seronegative individuals ($V$ and $R$ compartments in Fig. 1) can still be protected. Hence, seropositivity is assumed to be sufficient but not necessary for protection against infection, consistent with the fact that anti-PT IgG antibodies are not a serological correlate of protection[23]. We also note that seropositivity due to recent vaccination is ignored in this model. This assumption is justified because recently vaccinated individuals are generally excluded from serosurveys to limit the risk of false positives from causes different from exposure or infection[75].

By design, this model produced serological endpoints—such as seroprevalence and sero-incidence rates—comparable to serosurveys. In the following, we focus only on seroprevalence, the endpoint most typically estimated in these surveys (see Fig. 1 for the model-based definition). However, because we assume that seropositivity lasts, on average, one year (see below), the seroprevalence approximately equals the yearly sero-incidence rate in our model (based on the approximate formula: prevalence = incidence * duration). To assess whether seropositivity can reliably indicate a recent infection, we further calculated the PPV of seropositivity, which is defined here as the conditional probability of recent infection given seropositivity. Equivalently, the PPV represents the proportion of true cases among seropositive cases in serosurveys: PPV = true positives/(true positives + false positives), ranging from 0 (all seropositives result from immune boosting) to 1 (all seropositives result from a transmissible infection).

Hence, this model allowed us to parsimoniously capture and study the interpretation problem described in the introduction: without other information, seroprevalence includes both recent transmissible infections and immune boosts; consequently, seropositivity does not always indicate a recent infection, which may lead to false positives and a low PPV of serology for a given prevalence of actual infections.

## Model parametrization

**Fixed model parameters.** The model was structured by age, dividing individuals into two age groups during the first year of life and into one-year age groups from ages 1 to 79, resulting in a total of 81 age groups. The first age group represented newborns aged <2 or <3 months, reflecting current pertussis vaccination schedules that recommend administering the first vaccine dose a few months after birth[5]. Contacts between age groups were parameterized using SCMs derived from the work of Mistry et al.[46]. These SCMs were available in 35 countries worldwide and provided an age resolution of one year from age 0 to 79, enabling us to accurately capture contact patterns among infants and preschool children. Additionally, the model incorporated country-specific population age structures based on demographic data from 2010, as provided by Mistry et al.[46].

The main transmission parameters were fixed based on earlier modeling studies[64,65]. These parameters included the average latent and infectious periods and the relative transmissibility of secondary infections (Table 1). Importantly, since we focused exclusively on infection endpoints, we did not add an observation model to link our model outputs to reported disease cases.

Some serological parameters were assumed to be known and fixed according to a previous modeling study of the kinetics of anti-PT IgG after a lab-confirmed infection [5]. Specifically, we used the posterior distribution from the best-fitting power decay model described by Teunis et al.[26] to estimate the average time to seroconversion after exposure ($t_p = 23$ days) and the average duration of seropositivity ($t_n = 1$ year), both defined for a seropositivity threshold of 100 IU/mL.

**Estimated model parameters.** In contrast to the fixed parameters described above, the immune-boosting coefficient and the duration of protection were assumed to be unknown. This assumption was justified because of structural differences between our model and previous immune-boosting models[31,50,51,56], which prevented us from directly incorporating estimates from earlier studies. For example, though both our model and Lavine et al.'s model[31,51] stratify protection into a boosted and waning stage, the duration spent in the boosted stage is shorter in our model (because it represents the duration of seropositivity). As a result, using the estimates from Lavine et al. would lead to a higher force of infection in our model, requiring parametric adjustment (e.g., by increasing immune boosting or extending the duration of protection) to ensure comparability between the two models. In addition, there remains considerable uncertainty in the estimates of the immune-boosting coefficient, which were estimated at $\geq 10$ in Lavine et al.[31] and 6.6 (0.6–66) in Lavine et al.[51].

To simplify our analysis, we further assumed that infection- and wP-derived immunities had identical properties in terms of immune boosting ($\rho_R = \rho_V = \rho$) and the average duration of protection ($\alpha_R^{-1} = \alpha_V^{-1} = \alpha^{-1}$). This assumption is supported by immunological evidence showing that both natural infection and wP vaccination elicit a comparable immune response[59,60], as well as epidemiological evidence from modeling studies[56,57].

To estimate these two unknown parameters, we fitted our model to seroprevalence data from two large serosurveys in Europe[44,45]. We focused on these two studies because their results were standardized to ensure comparability of the serological assays across the participating countries. The first study, conducted in six countries in the mid −1990s, covered all age groups and reported seroprevalence based on a seropositivity threshold of 125 IU/mL[44]. The second study, conducted in fourteen countries during the early 2010s, focused only on adults aged 20–39 and reported seroprevalence based on a seropositivity threshold of 100 IU/mL[45].

To include countries in our study, we applied three criteria:
1. Social contact data available from Mistry et al.[46]
2. Stable high (> 80%) vaccine coverage from the start of wP vaccination until the collection of serological samples.
3. Switch to aP vaccines for primary immunization ≤5 years before the collection of serological samples.

These criteria, thus, restricted our analysis to countries with mature and stable vaccination programs before or very shortly after the switch to aP vaccines. In particular, the last criterion was applied to simplify our model and focus our analysis on infection/wP-derived protection, which differs from aP-derived protection[61,62]. Of note, we did not consider a more recent European serosurvey[42] because all participating countries had switched to aP vaccines >5 years before sample collection.

In each selected country, we ran 10 stochastic simulations from the beginning of wP vaccination until the serological survey period. We then calculated the model-predicted seroprevalence in adults aged

20–44 (for countries from ref. 44) or 20–39 (for countries from ref. 45). For model-data comparison, we used weighted linear regression with the observed seroprevalence ($S_P^{(obs)}$) as the outcome and the predicted seroprevalence ($S_P^{(mod)}$) as an offset, with weights equal to the inverse variance of the observed seroprevalence estimates. Such weighting allowed us to account for the statistical uncertainty due to the limited sample size of the serosurveys. Mathematically, this regression model is represented by the equation:

$$S_P^{(obs)} = S_P^{(mod)} - \alpha \qquad (1)$$

In this regression model, the parameter $\alpha$ thus represented the MWE between the model predictions and the data.

We repeated this procedure for multiple pairs of the immune-boosting coefficient and average duration of protection ($\rho, \alpha^{-1}$). Specifically, we considered four values of immune-boosting (0.5, 1, 2, 5) and multiple average durations of protection ranging from 10 to 90 years (by increments of 10 years). Of note, in the absence of immune boosting ($\rho = 0$), the duration of vaccine-derived protection follows an exponential distribution (with rate $\alpha$) and the duration of infection-derived protection a generalized Erlang distribution (a sum of two exponential distributions with rates $1/t_n$ and $\alpha$). In both cases, the duration of protection is inherently variable, with a large fraction of individuals losing protection before the average duration (63% for the exponential distribution). Any pair resulting in a non-significant MWE (i.e., not significantly different from 0, according to a t-test) was considered an admissible estimate consistent with the data. For a given immune-boosting level, we calculated an approximate 95% confidence interval for the average duration of protection as the range of all admissible estimates. This approach is based on the connection between confidence intervals and hypothesis testing, such that a 95% confidence interval can be calculated as the set of null hypotheses that are not rejected at level 5% (ref. 76, chapter 5).

**Model predictions across representative countries**
**Next generation matrix clustering.** To get a broader picture of the potential shortcomings of serology, we predicted the seroprevalence and PPV of seropositivity from serosurveys across various countries worldwide. To simplify our analysis, we did not consider all 35 countries with social contact data available from Mistry et al.[46]. Instead, we clustered the NGMs to identify groups of countries with broadly similar transmission dynamics.

In every country, we first calculated the NGM and the corresponding basic reproduction number ($R_O$, defined as the leading eigenvalue of the NGM)[77]. Next, we computed the pairwise Manhattan distance between every pair of country-level NGMs to create a dissimilarity matrix encompassing all countries. Finally, we applied agglomerative nested hierarchical clustering and used the silhouette method to determine the optimal number of clusters[78]. For every cluster with ≥2 countries, we selected the country with the largest population as the representative for that cluster.

**Simulation protocol.** In every representative country identified through NGM clustering, we simulated our serotransmission model for 150 years to reach equilibrium in the prevaccine era and another 150 years in the vaccine era. We ran ten replicate stochastic simulations and recorded the seroprevalence in three adult age groups—20–39, 40–59, and 60–79—for the last twenty years of the simulated period (200 simulation-years in each age group). We considered these specific age groups because serosurveys often focus on adults[42,45], in whom pertussis is less likely to be reported.

## Numerical implementation and code availability statement
The serotransmission model was developed using the pomp package (version 5.10)[79,80] in R version 4.4.1[81]. The NGM clustering was performed using the R package clValid[82], with visualization done using the package factoextra[83].

## Reporting summary
Further information on research design is available in the Nature Portfolio Reporting Summary linked to this article.

## Data availability
All data are available on GitHub (https://github.com/DomenechLab/Pertussis_seroprevalence) and Zenodo[84].

## Code availability
All programming codes are available on GitHub (https://github.com/DomenechLab/Pertussis_seroprevalence) and Zenodo[84].

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

## Acknowledgements
We thank Peter Teunis for sharing codes and data to estimate the serological parameters and Christian Gunning for helpful comments on the manuscript. This study was funded by the Max Planck Society.

## Author contributions
M.D.d.C conceived of the study design, implemented the model, and checked data extraction from seroprevalence studies. A.W. conceived of the study design, conducted the review of seroprevalence studies, and checked the model codes for reproducibility. T.D. provided content expertise. P.R. oversaw the analysis. All authors helped draft and approved the final version of the manuscript.

## Funding

## Competing interests
P.R. received funding from Sanofi for a research project on pertussis vaccines. A.W. received consulting fees from Vaxcyte for work unrelated to pertussis. All other authors declare no competing interests related to this study.
