## [Transparent Peer Review file · Nature Communications]

Natural immune boosting biases pertussis infection estimates in seroprevalence studies

Corresponding Author: Dr Matthieu Domenech de Cellès

Version 0:

Reviewer comments:

Reviewer #1

(Remarks to the Author)

de Cellès et al present their study on assessing the reliability of serosurveys accounting for the impact immune boosting might have on accurately estimating pertussis seroprevalence in different age groups. The work is of significance to the field and the transmission model structure inclusive of serological components and the quantification of immune boosting might play an important role in future pertussis serosurveillance studies. The methodologies are well detailed and may be reproducible.

Major comments

1. It is understood that the model was fit to data from two large European serosurveys to estimate durations of infection. Could you report any model fit parameters?
2. It seems that this model assumes that from the vaccinated state you would either have vaccine failure which moves you to S2 (though I do not see a parameter for that) or eventually to VE when exposed. After being exposed, you can only move to the boosted compartment (Vp), without provisions for becoming infected rather than exposed. How does this model then capture waning immunity after vaccination? Presumably some individuals move from V to S2 capturing waning as well as vaccine failure, but that doesn't seem to be part of the parameterization. Please clarify why there is no movement from Ve to infected. It seems like a parameter for vaccine failure would also be warranted.
3. What about the different age structures of the 12 countries selected. India and China of course have a much different age pyramid compared to the European countries. How could this affect your PPV estimates? It is noted in your limitations that the immunity did not vary by age, so this also interacts with the fact that there are different age structures in these countries.
4. The introduction is long and makes it hard to focus on the major aim of the paper. Consider cutting down the literature review part from line 113 – 133. Replace with a short summary of the problem and reference some of these studies.
5. Some parts of the results section are missing comparing and contrasting the study findings in a quantitative manner when exploring different parameters (e.g. ρ) by age group. There is a lot of increase or decrease without quantifying these changes and/or magnitude. For example, line 395-396 "In a given age group, the force of infection was predicted to decrease as the strength of immune boosting increased". By how much? Are these changes significant?

Minor comments

1. Line 51- you present pertussis death estimates from over 10 years ago. Have there not been any updated estimates in the literature?
2. Line 55-56—Some discussion or comment on the potential reason for these large epidemics following the pandemic would be helpful to the reader.
3. Introduction, line 97 – remove space before "We believe this definition..."
4. Line 228-229—include the author last names like you did in line 225 (lavine et al) or just the numbered reference. Do not include "Ref."
5. Line 249—criteria for programs to have switched to aP less than 5 years before samples collected. You describe how this is a criteria for country inclusion but do not provide an explanation for why this is a criteria or how the difference could affect your model. In the introduction you briefly note that countries have switched, but do not mention the differences between the vaccines. If this is a key piece to the study, it would be good to provide the reader the context of why this was an important criteria. I see it noted in the limitations, so perhaps just an earlier brief explanation.
6. You may want to include figure and/or table references in results (from line 361 – 365). It's challenging to visualize the findings without referencing them.

7. In lines 381-389 you refer to different panels of figure 4, e.g. Fig. 4A, but the figure is not labelled as such. Consider labeling your panels a-f and also label other figures similarly. I think this is a requirement in Nature communications and the likes.
8. Remove space in line 491.
9. Remove other on line 546
10. Dendrogram is misspelled in the supplement Figure S2.

(Remarks on code availability)

Reviewer #2

(Remarks to the Author)

(Remarks on code availability)

The code is well written and sufficiently annotated. The example model simulation code was successfully run, and the results were reproducible.

Reviewer #3

(Remarks to the Author)

Dear authors

Thank you for the opportunity to review your manuscript. Your work is a vital contribution to the field of pertussis epidemiology. You point to a well-known, yet still not broadly acknowledged, let alone understood or quantified limitation of sero-epidemiologic studies of pertussis infection. I certainly understand and agree with the concept that some exposures to *B. pertussis* may trigger an immune stimulation in primed individuals without necessarily resulting in a transmissible infection. That this implies that serosurveys should not be interpreted as accurate measures of transmissible infections is well explained and your model appears to effectively demonstrate that. I am not convinced, though, that this is not epidemiologically relevant as serosurveys would still reflect the level of exposure (or circulation) and thus one element of the force of infection. The fact that some exposures do not result in transmissible infection may be more related to the other element of the force of infection: susceptibility. It is unclear to me from your manuscript if you factored in the model a variation of the probability of immune boosting over time (variable ρ) as protection from transmissible infection wanes. For this reason, my main comment about your manuscript is that I believe the inability of your model to produce a quantification for the immune boosting coefficient makes me question the validity of estimating a PPV for serosurveys.

Here a few complementary comments by lines:

LINES 82-87: True, but the several-fold higher incidence of pertussis than notified in surveillance systems is also supported by healthcare claims data which found that diagnosed pertussis cases are generally several orders of magnitude higher than surveillance data suggest. One may question the specificity of healthcare claims data since the vast majority of these cases are not laboratory confirmed. Yet, pertussis is largely under-recognized and under-diagnosed, so while being imperfect, clinical diagnosis of pertussis in these databases may provide a closer estimate of the incidence of disease incidence than surveillance data. Therefore, all measures being imperfect, the reality likely sits somewhere between serosurveys, healthcare claims and surveillance data estimates.

LINE 102: "without culture positivity"... the well known lack of sensitivity of culture challenges the reliance on absence of culture positivity to support the hypothesis of non-transmissible immune boost

LINES 102-105: The natural infectious dose for *Bp* is unknown and therefore the absence of shedding detected in this study is difficult to interpret from a transmission point of view.

LINES 105-108: yes, this study indicates that single-sample serology lacks specificity in detecting clinical cases of pertussis but it does not say anything about transmissibility of infection. Even asymptomatic infections may be transmissible.

LINE 158: anti-PT IgG... Is using PT as the only IgG of reference sufficient to consider immune boosting? Do we know enough about the dynamics of production of PT (and the resulting immune stimulation ag. PT) during the course of infection to consider that anti-PT IgG as the correct and sole marker of infection and immune boosting? Would anti PRN or FHA IgG or other antigens help with that?

LINES 166-167: for $\rho < 1$, an 167 exposure sufficient for infection is less likely to induce immune boosting... how do you conceive that an exposure sufficient to induce infection would not stimulate the immune system?

LINE 216: ($t_p = 23$ days)... This does not appear consistent with the data from the human challenge studies. I believe they found that exposure can result in detection of IgG secreting cells in as little as 14 days

LINES 237-243: the calibration of your model relied on cross-sectional surveys across age groups. This may introduce

unknown biases as these data reflect but do not account for the diverse histories of prior exposures of the different age groups to *B. pertussis*, which would have varied over the years.

LINES 247-248: Why this criterion? If the model is robust enough, wouldn't it be possible to run it with lower wP VCR such as West Germany and Italy?

LINE 270: duration of immunity... since you are dealing with immune marker evaluation the term "immunity" as used here can be confusing. It may be preferable to be specific and talk about "protection"

LINES 273-275: This is an important caveat on the results you present. I recommend it be clarified again when presenting and discussing results on duration of protection

LINES 326-333: I'm not sure I follow the reasoning. You found that the modelled seroprevalence fitted the observed seroprevalence rather closely irrespective of the rho parameter. Of course PPV being derived from rho, the higher rho the lower the PPV. But does that really mean that large parts or a majority of seropositivity results from immune boosting? The disease surveillance at the time the serosurveys you used were conducted was very insensitive with lack of sensitivity increasing with age, and it is largely the case today too. Wouldn't your results (fitness of the model) then suggest instead that pertussis seroprevalence trends/distributions are indicators of force of infection (or rather of circulation) and parallel rather than accurately measure disease incidence?

LINES 470-475: similarly, while I agree with your conclusion that immune boosting is a factor to consider as limitation in quantifying transmissible infection from seroprevalence studies, I'm not sure it is valid to estimate PPV since rho could not be estimated.

LINES 477-485: I can certainly agree that comparing case notification data and seroprevalence data is fraught with biases. However, for the same reasons, it is as inadequate to discount the evidence of serosurveys vs case surveillance. Case surveillance is well documented to under-estimate disease burden, and completely misrepresents the dynamics of transmissible infections. Both data are interesting in that they reflect, albeit imperfectly, different aspects of the pathogen dynamics: circulation vs. disease

LINES 494-495: While the skewing of the immune response is comparable between natural infection and wP vaccines, I'm not sure it is correct to say the immune response is similar as one is led by mucosal stimulation whereas the other is parenteral. In humans, data is difficult to generate/interpret data on this topic. The baboons challenge data do suggest a difference in effect on colonization between convalescent and wP vaccinated baboons.

LINES 498-499: What about the interplay between a differential effect on transmission and observed duration of protection? Is it possible that difference in immune response between aP and wP vaccines may lead to a differential herd effect that could translate at population level into a different force of infection and thus an apparent difference in duration of protection, without there necessarily being one. Could your model help explore this?

(Remarks on code availability)

Reviewer #4

(Remarks to the Author)

The article presents a model to assess the reliability of seropositivity as a marker for pertussis infection. It distinguishes between seropositivity, symptomatic and asymptomatic infections, and highlights immune boosting—exposure triggering a detectable response without transmission—as a source of overestimation in serosurveys. The authors address these ambiguities in the literature through a literature review. Then, the authors propose a model that integrates infection, immune boosting, and seropositivity dynamics, accounting for anti-pertussis toxin IgG titers, as well as demographic and social factors. Calibrated with late whole-cell vaccine era data from six European countries, the model reveals a weak correlation between infection and seropositivity, reflected in a low positive predictive value (PPV), particularly among younger populations likely due to routine vaccine boosters and higher contact rates that favor transmission. Waning immunity and social contact patterns also impact reliability, underscoring the need for refined tools to interpret seroprevalence data. The article is well written and in general clear. Several key questions emerge:

1. Regarding the literature review, the authors should provide a clearer description of the methodology used, including the search strategy, whether study selection was performed by one or multiple reviewers, data extraction, and quality assessment. Additionally, they should clarify how they adhered to the PRISMA-P checklist, and I recommend including the checklist as supplementary material."
2. More detailed explanations are needed regarding the definitions used—such as exposure, immune boosting, and seropositivity—and the literature supporting these distinctions should be cited.
3. The model was calibrated using a survey from the late wP vaccine era rather than a more recent one, which raises several questions:
 - How realistic is the assumption that vaccine-derived immunity, particularly from whole-cell vaccines, is imperfect and may result in immediate failure (primary vaccine failure), followed by waning immunity mentioned in the Introduction?
 - How might the findings differ in the acellular pertussis (aP) vaccine era, and how can the model be applied in countries using the aP vaccines (most countries now)?
4. The cyclical pattern as well as seasonality of pertussis may influence seroprevalence—how was this accounted for in the

model? Additionally, were differences in vaccine booster policies across the studied countries considered?

5. Can the model integrate other factors such as social changes in the population, non-pharmaceutical mitigation measures, macrolide resistance etc.?

6. The authors should specify how the model could be used, for which populations it would be most reliable, and how it could be integrated with other sources to identify transmission.

(Remarks on code availability)

Version 1:

Reviewer comments:

Reviewer #1

(Remarks to the Author)

The authors have addressed all of my comments adequately.

(Remarks on code availability)

Reviewer #2

(Remarks to the Author)

(Remarks on code availability)

The code was reviewed for overall reproducibility during the first round of review. It was well written, reproducible and will prove useful to the field.

Reviewer #3

(Remarks to the Author)

Thank you for your careful consideration of my comments and for your comprehensive explanations. With the revisions you made following my and other reviewers' comments, I believe your manuscript presents a very valuable new way to look at pertussis epidemiological evidence. I hope this can contribute to decision making in vaccination policies in the future.

(Remarks on code availability)

Reviewer #4

(Remarks to the Author)

(Remarks on code availability)

We thank our four reviewers for their thorough comments and valuable suggestions. Please find below a point-by-point response to every reviewer's comment.

Reviewer #1 (Remarks to the Author): de Cellès et al present their study on assessing the reliability of serosurveys accounting for the impact immune boosting might have on accurately estimating pertussis seroprevalence in different age groups. The work is of significance to the field and the transmission model structure inclusive of serological components and the quantification of immune boosting might play an important role in future pertussis serosurveillance studies. The methodologies are well detailed and may be reproducible.

Thank you.

Major comments

1. It is understood that the model was fit to data from two large European serosurveys to estimate durations of infection. Could you report any model fit parameters?

The model was indeed fit to seroprevalence data from two European serosurveys. Specifically, we estimated the average duration of infection and wP-derived immunity (parameter α^{-1}) while varying the strength of immune boosting (parameter ρ). The corresponding estimates are reported in Table 2 in the main text, reproduced below for the reviewer's convenience.

Fixed value of boosting coefficient (ρ)	Estimate (95% confidence interval) of the average duration of infection/wP-derived immunity (α^{-1}), years	MWE, %	PPV range, %
0.5	30 (20–40)	–0.4	47–62
1.0	30 (30–50)	0.0	29–41
2.0	40 (30–60)	–0.1	16–27
5.0	50 (40–80)	0.1	6–12

Table 2

2. It seems that this model assumes that from the vaccinated state you would either have vaccine failure which moves you to S2 (though I do not see a parameter for that) or eventually to VE when exposed. After being exposed, you can only move to the boosted

compartment (Vp), without provisions for becoming infected rather than exposed. How does this model then capture waning immunity after vaccination? Presumably some individuals move from V to S2 capturing waning as well as vaccine failure, but that doesn't seem to be part of the parameterization. Please clarify why there is no movement from Ve to infected. It seems like a parameter for vaccine failure would also be warranted.

The reviewer is exactly right: as explained in Methods/Model structure, we assumed two modes of failure for vaccine-derived immunity (based on previous fits in Massachusetts, USA [1]): primary vaccine failure and waning immunity. For simplicity, primary vaccine failure was not explicitly represented in the model diagram; however, it is captured through the effective vaccination coverage p_v . Specifically, $p_v = P_v(1 - \epsilon_v)$, where $P_v = 0.95$ is the assumed (total) vaccination coverage and $\epsilon_v = 0.05$ the probability of primary vaccine failure. As a result, $p_v \approx 0.9$, as indicated in Table 1; we then calculated the effective vaccination rate ν from the effective vaccination coverage p_v . We have updated Table 1 to clarify this point.

Regarding the lack of movement from V to infected, this reflects our assumption that vaccine-derived immunity is not leaky. Again, this assumption is based on our study in Massachusetts, USA, where we specifically tested the leakiness hypothesis but found no evidence for it.

3. What about the different age structures of the 12 countries selected. India and China of course have a much different age pyramid compared to the European countries. How could this affect your PPV estimates? It is noted in your limitations that the immunity did not vary by age, so this also interacts with the fact that there are different age structures in these countries.

The reviewer raises an important point. In response, we created a new figure (Figure S4, reproduced below) comparing the population structure of the sixteen countries considered in our analysis. As intuited by the reviewer, India has a distinct age pyramid, similar to that of Israel. However, these countries do not appear as clear outliers in Figure 4 (except, perhaps, in the high-boosting scenario, where the seroprevalence predicted for 20–39 years is somewhat lower compared to the other countries). In our model, variations in transmission dynamics across countries are determined not only by differences in demographic structures but also by variations in social contact matrices (Figure S3). As a result, it is challenging to isolate the effect of demography on PPV in our simulations. However, we acknowledge that this would indeed be an interesting line of future research.

Figure S4

4. The introduction is long and makes it hard to focus on the major aim of the paper. Consider cutting down the literature review part from line 113 – 133. Replace with a short summary of the problem and reference some of these studies.

We appreciate this suggestion, and in response, we have now relocated the presentation and discussion of this review to the Methods and Results sections. As a result, the introduction is now shorter (6 paragraphs) and more streamlined.

5. Some parts of the results section are missing comparing and contrasting the study findings in a quantitative manner when exploring different parameters (e.g. ρ) by age group. There is a lot of increase or decrease without quantifying these changes and/or magnitude. For example, line 395- 396 “In a given age group, the force of infection was predicted to decrease as the strength of immune boosting increased”. By how much? Are these changes significant?

In response, we have added text to quantify these changes. Note that we do not report whether these changes are statistically significant; indeed, the significance level is arbitrary in a simulation study, as any amount of data can be simulated.

Minor comments

1. Line 51- you present pertussis death estimates from over 10 years ago. Have there not been any updated estimates in the literature?

This is an excellent suggestion. We updated this figure based on the most recent pre-pandemic data (19.9 million new cases in children aged 0–14 in 2019) from the Global Burden of Disease study [2].

2. Line 55-56—Some discussion or comment on the potential reason for these large epidemics following the pandemic would be helpful to the reader.

To the best of our knowledge, the causes of these post-pandemic resurgences have not yet been fully elucidated. Based on epidemic theory, a plausible hypothesis is the gradual build-up of susceptibles due to reduced circulation of pertussis during the early COVID-19 pandemic (as non-pharmaceutical interventions against COVID-19 had non-specific effects on other infectious diseases, including pertussis and other bacterial diseases [3]). However, we believe it is too early to tell, so we prefer not to comment on that.

3. Introduction, line 97 – remove space before “We believe this definition...”

Corrected, thanks.

4. Line 228-229—include the author last names like you did in line 225 (lavine et al) or just the numbered reference. Do not include “Ref.”

Corrected as suggested (first option).

5. Line 249—criteria for programs to have switched to aP less than 5 years before samples collected. You describe how this is a criteria for country inclusion but do not provide an explanation for they this is a criteria or how the difference could affect your model. In the introduction you briefly note that countries have switched, but do not mention the differences between the vaccines. If this is a key piece to the study, it would be good to provide the reader the context of why this was an important criteria. I see it noted in the limitations, so perhaps just an earlier brief explanation.

We added the following sentence to clarify this point:

“In particular, the last criterion was applied to simplify our model and focus our analysis on infection/wP-derived immunity, which differs from aP-derived immunity [4,5].”

6. You may want to include figure and/or table references in results (from line 361 – 365). It’s challenging to visualize the findings without referencing them.

In response, we now reference specific panels of Figure 4 throughout the results.

7. In lines 381-389 you refer to different panels of figure 4, e.g. Fig. 4A, but the figure is not labelled as such. Consider labeling your panels a-f and also label other figures similarly. I think this is a requirement in Nature communications and the likes.

In response, we now explicitly label A–F the different panels of Figure 4.

8. Remove space in line 491.

Corrected, thanks.

9. Remove other on line 546

Corrected, thanks.

10. Dendrogram is misspelled in the supplement Figure S2.

The legend now includes the correct spelling (dendrogram).

Reviewer #2 (Remarks to the Author):

Reviewer #2 (Remarks on code availability): The code is well written and sufficiently annotated. The example model simulation code was successfully run, and the results were reproducible.

Thank you. Please note that we have updated the GitHub repository before resubmission. In particular, it now contains a new script (“m-run_simulations_alt_model_homogeneous.R”) that runs a simulation study for an alternative formulation of our model, with two boosting parameters (see our reply to the first comment of Reviewer 3 below).

Reviewer #3 (Remarks to the Author):

Dear authors Thank you for the opportunity to review your manuscript. Your work is a vital contribution to the field of pertussis epidemiology. You point to a well-known, yet still not broadly acknowledged, let alone understood or quantified limitation of sero-epidemiologic studies of pertussis infection. I certainly understand and agree with the concept that some exposures to *B. pertussis* may trigger an immune stimulation in primed individuals without necessarily resulting in a transmissible infection. That this implies that serosurveys should not be interpreted as accurate measures of transmissible infections is well explained and your model appears to effectively demonstrate that. I am not convinced, though, that this is not epidemiologically relevant as serosurveys would still reflect the level of exposure (or circulation) and thus one element of the force of infection. The fact that some exposures do not result in transmissible infection may be more related to the other element of the force of infection: susceptibility. It is unclear to me from your manuscript if you factored in the model a variation of the probability of immune boosting over time (variable ρ) as protection from transmissible infection wanes. For this reason, my main comment about your manuscript is that I believe the inability of your model to produce a quantification for the immune boosting coefficient makes me question the validity of estimating a PPV for serosurveys.

Thank you for your encouraging comments. We agree that serosurveys reflect the level of exposure, a point we discussed in the initial submission (Discussion, fifth paragraph): “More broadly, our study emphasizes the key difference between exposure and infection, as seropositivity always indicates the former but not necessarily the latter.” We revised the abstract to clarify this point: “Thus, although serosurveys are unquestionably useful for quantifying pertussis exposure levels, the common interpretation of seroprevalence as a measure of recent infections may lead to underestimating the impact of pertussis vaccines.”

Regarding the probability of immune boosting varying over time, the reviewer raises an important point. Our base model was derived from the model developed by Lavine *et al.* [6,7], which models immune boosting in arguably the simplest way possible. Hence, we agree with the reviewer that, in practice, immune boosting is likely more complex, with a continuous variation in the probability of immune boosting over time since infection or vaccination. As is often the case in mathematical modeling, models must strike a balance between biological realism and mathematical tractability, particularly in terms of parameter identifiability, which was already problematic for the simplest model. Nevertheless, the reviewer’s comment is well-taken. In response, we considered an alternative model with a more precise description of immune boosting represented by the following schematic (see new Figure S6):

Figure S6

In this extended model, vaccine- and infection-derived immunities are stratified into two stages (V_1, V_2 and R_1, R_2 , respectively), with a boosting strength of ρ during the first stage and $\kappa\rho$ during the second. Hence, as suggested by the reviewer, this model captures a gradual decline (if $\kappa \leq 1$) in the strength of immune boosting.

For simplicity, we omitted the compartments representing exposed states (V_E and R_E) and secondary infections (E_2 and I_2) and ignored age structure (homogeneous model). As before, we used a grid search to estimate waning rate values (parameter α) consistent with observed seroprevalences. We considered two target levels of overall seroprevalence, based on two nationwide serosurveys that included all age groups: 10% (Netherlands, 2006–2007 [8]) and 20% (Australia, 1997–1998 [9]). Because both studies reported seroprevalence estimates for an anti-PT IgG cut-off of 62.5 IU/mL, we fixed the average duration of seropositivity accordingly ($t_n = 1.9$ years). To our knowledge, these two estimates are the highest reported in the vaccine era. We repeated the estimation for two values of ρ (0.5 and 1), which were the most unfavourable to our central hypothesis that serosurveys have low PPV (see Tables 2–3). For completeness, we also repeated the analysis for the homogeneous version of the base model (Fig. S5, top panel). For both models, we assumed a basic reproduction number of 15 and fixed other parameters as in Table 1.

As shown in the new Table S7 (reproduced below for the reviewer’s convenience), our main results were robust: a combination of low waning immunity and low PPV best explained the target seroprevalence in every scenario. For the homogeneous version of the base model and a target seroprevalence of 20%, the estimated average duration of immunity ranged from 60 years ($\rho = 0.5$) to 150 years ($\rho = 1$). Again, we note the underlying exponential distribution for the duration of vaccine-derived immunity (without boosting), such that these averages translate into a sizeable fraction of vaccinees losing immunity within their lifespan [1,10]. Because of the different distribution assumed for the alternative model (Gamma instead of Exponential), the corresponding estimates were lower but still high, ranging from 30 years ($\rho = 0.5$) to 50–60 years ($\rho = 1$). In keeping with the results from Tables 2–3, the PPV of seropositivity in serosurveys increased as the boosting coefficients decreased but

remained low in all scenarios (overall range: 8–25%). Hence, these additional simulations suggest that our main results are relatively unaffected by different modeling assumptions regarding waning immunity and boosting. These new analyses are now reported in the supplement (Text S1.3, Figure S6, and Table S7) and the main text (Results/Alternative model structure and parametrization of immune boosting).

Model	Target seroprevalence	ρ (fixed)	κ (fixed)	α^{-1} (estimated, years)	PPV (%)
Base (Fig. S5, top schematic)	10%	0.5	—	250	14
		1		>500	<8
	20%	0.5		60	15
		1		150	8
Extended (Fig. S5, bottom schematic)	10%	0.5	0	90	21
			0.5	100	15
			1	95	14
		1	0	100	12
			0.5	>500	<8
			1	>500	<7
	20%	0.5	0	30	25
			0.5	30	18
			1	30	15
		1	0	60	14
			0.5	55	9
			1	50	8

Table S7

Here a few complementary comments by lines:

LINES 82-87: True, but the several-fold higher incidence of pertussis than notified in surveillance systems is also supported by healthcare claims data which found that diagnosed pertussis cases are generally several orders of magnitude higher than surveillance data suggest. One may question the specificity of healthcare claims data since the vast majority of these cases are not laboratory confirmed. Yet, pertussis is largely under-recognized and under-diagnosed, so while being imperfect, clinical diagnosis of pertussis in these databases

may provide a closer estimate of the incidence of disease incidence than surveillance data. Therefore, all measures being imperfect, the reality likely sits somewhere between serosurveys, healthcare claims and surveillance data estimates.

We agree with this comment, and we discuss the shortcomings of case notification data in the introduction: “These factors collectively contribute to case underreporting, which is estimated to be substantial for pertussis.”

We also agree that reality must sit somewhere in the middle. Hence, the way forward is to combine information from all available data sources, which is made possible by our serotransmission model. We develop this idea in the conclusion: “Hence, our model—or variations thereof—may prove useful for analyzing seroprevalence data and synthesizing evidence from other sources, including case notification data. Eventually, fitting such models to all available data will improve our estimates of pertussis infection rates and help resolve the ongoing disagreements within the field.”

LINE 102: "without culture positivity"... the well known lack of sensitivity of culture challenges the reliance on absence of culture positivity to support the hypothesis of non-transmissible immune boost

We note that the sensitivity of culture can be quite high if performed during the first two weeks after cough onset (e.g., 74–93% according to [11]), although it indeed drops sharply thereafter. More broadly, according to official guidelines from the CDC¹, culture remains the gold standard for pertussis diagnosis during the first two weeks of illness. Regarding the study in Sweden that we cited [12], active surveillance of pertussis cases in households was conducted, such that any positive culture in a household member triggered diagnostic procedures (including culture) in all other household members. In this context, one expects all cases to have been tested and diagnosed early, leading to increased sensitivity of culture. Altogether, we believe citing this study as empirical evidence of immune boosting is legitimate.

LINES 102-105: The natural infectious dose for Bp is unknown and therefore the absence of shedding detected in this study is difficult to interpret from a transmission point of view.

We agree with the reviewer that the natural infectious dose of Bp is unknown. (Although we note that de Graaf et al. [13] mention an admittedly small and old study: “In a very small pediatric study carried out in 1933, Bp disease was induced by exposing 2 presumably immunologically naive children to 140 cfu of Bp”.) Yet the finding that 100,000 CFU causes seroconversion without shedding is direct empirical evidence that at least some level of exposure can cause immune boosting. Hence, we believe the results of this study are relevant and legitimate for the discussion of immune boosting we develop in this paragraph.

¹ <https://www.cdc.gov/pertussis/php/laboratories/index.html>

LINES 105-108: yes, this study indicates that single-sample serology lacks specificity in detecting clinical cases of pertussis but it does not say anything about transmissibility of infection. Even asymptomatic infections may be transmissible.

We agree that this study does not provide direct information on transmissibility, and we did not specifically address this point in the context of this study. Instead, we reported the authors' conclusion about case misclassification and poor diagnostic specificity of serology in their setting. Unlike the other two studies cited in this paragraph, this study does not provide direct evidence of immune boosting; however, its results are consistent with immune boosting. We have revised this sentence to clarify this point: "[...] the authors interpreted this discrepancy as evidence of case misclassification and poor diagnostic specificity of serology in their setting, suggestive of immune boosting."

LINE 158: anti-PT IgG... Is using PT as the only IgG of reference sufficient to consider immune boosting? Do we know enough about the dynamics of production of PT (and the resulting immune stimulation ag. PT) during the course of infection to consider that anti-PT IgG as the correct and sole marker of infection and immune boosting? Would anti PRN or FHA IgG or other antigens help with that?

Although *B. pertussis* exposure or infection triggers an immune response targeting multiple antigens, PT is the only antigen specific to *B. pertussis*, and cross-reactive responses following infection with other *Bordetella* species are possible for other antigens, such as FHA and PRN. As a result, most serosurveys focus only on this antigen, which explains our choice to model the seroprevalence of anti-PT IgG. We note that the kinetics of anti-PT IgG are relatively well characterized from the work of Peter Teunis (and others), and we built on this work to fix the serological parameters in our model (see Table 1).

LINES 166-167: for $\rho < 1$, an exposure sufficient for infection is less likely to induce immune boosting... how do you conceive that an exposure sufficient to induce infection would not stimulate the immune system?

We agree this is a very conservative assumption, as a value of $\rho < 1$ is indeed unlikely based on immunological reasoning. Nevertheless, we considered a scenario where $\rho = 0.5$ based on previous estimates from Lavine et al. (e.g., 95% CI of 0.66–6.6 in Ref. [7]). We note this scenario is the most unfavourable to our working hypothesis. Hence, discarding this scenario would make our main results—observed seroprevalence levels can be explained by a combination of low waning immunity and low PPV—stronger. We have added this sentence to emphasize this point (Methods/Model structure):

“For completeness, we considered the former, converse scenario of hyposensitive boosting, but we note this assumption is conservative and immunologically unlikely.”

LINE 216: ($t_p = 23$ days)... This does not appear consistent with the data from the human challenge studies. I believe they found that exposure can result in detection of IgG secreting cells in as little as 14 days.

This estimate was based on the best kinetic model of IgG decay from Teunis et al. [14], for an IgG seropositivity threshold of 100 IU/mL. Although this estimate may differ from that of the challenge study, we note that the difference is too small to have an impact on our predictions of seroprevalence, given the much longer timescale resulting from the assumed duration of seropositivity (1 year).

LINES 237-243: the calibration of your model relied on cross-sectional surveys across age groups. This may introduce unknown biases as these data reflect but do not account for the diverse histories of prior exposures of the different age groups to *B. pertussis*, which would have varied over the years.

The reviewer raises an important point. As our model is dynamical and age-structured, it accounts for differences in the history of prior exposures between age groups. In our model, the dynamical consequences of these differences will be determined by other factors such as waning immunity, immune boosting, age-specific contact patterns, susceptibility to infection, and force of infection. Hence, our model can partially account for this important source of bias in the data. However, as we discussed in the limitations, the model-data comparison is only approximate because we fit to a single cross-sectional seroprevalence estimate and do not incorporate other longitudinal information about past epidemics. We have revised the main text (Discussion, paragraph 8) to better acknowledge this limitation:

“In addition, we did not include other data to inform the model about the past epidemics and age-specific history of prior exposures and infections in each country. In practice, the serosurveys may have captured a different time of the epidemic cycle in every country, yet our model predictions were based on averages across multiple epidemic cycles (Fig. S5).”

LINES 247-248: Why this criterion? If the model is robust enough, wouldn't it be possible to run it with lower wP VCR such as West Germany and Italy?

The reviewer is correct that our model is flexible enough to incorporate temporal variations in vaccination coverage. However, we applied this criterion to select countries with stable vaccination, which allowed us to simplify the analysis and assume a single vaccination coverage during the simulated study period. The inclusion of countries with unstable coverage, although technically possible, would have required detailed time series of vaccination coverage; however, such data are generally unavailable or difficult to obtain.

LINE 270: duration of immunity... since you are dealing with immune marker evaluation the term "immunity" as used here can be confusing. It may be preferable to be specific and talk about "protection"

This is an excellent suggestion, and we have revised the text accordingly.

LINES 273-275: This is an important caveat on the results you present. I recommend it be clarified again when presenting and discussing results on duration of protection

The reviewer is exactly right: this is a critical assumption of our model. In the results, we paid careful attention to presenting not only estimates for the average duration of vaccine protection, but also other quantiles (Results/Estimation and model fit to seroprevalence data): “These averages translated into a sizeable fraction of 10–15% of vaccinees losing immunity within 5 years and 18–28% within 10 years.” In response to the reviewer, we further revised the text (especially the discussion) to emphasize that the estimated duration of protection, although long on average, was inherently variable. Please also note our new analysis based on an extended model (Fig. S6), in which we made a different assumption for this distribution (Gamma instead of Exponential). Encouragingly, our main results remained robust despite this important change (Table S7).

LINES 326-333: I'm not sure I follow the reasoning. You found that the modelled seroprevalence fitted the observed seroprevalence rather closely irrespective of the ρ parameter. Of course PPV being derived from ρ , the higher ρ the lower the PPV. But does that really mean that large parts or a majority of seropositivity results from immune boosting? The disease surveillance at the time the serosurveys you used were conducted was very insensitive with lack of sensitivity increasing with age, and it is largely the case today too. Wouldn't your results (fitness of the model) then suggest instead that pertussis seroprevalence trends/distributions are indicators of force of infection (or rather of circulation) and parallel rather than accurately measure disease incidence?

The reviewer is correct that the parameter ρ sensitively determines the PPV of serology. In our analyses, we considered a range of values for this parameter, including a scenario of hyposensitive boosting ($\rho = 0.5$), which, as discussed above in response to the reviewer's comment on lines 166–167, may be immunologically unlikely and too unfavorable to our working hypothesis. Thus, to the extent this range is realistic, we can conclude about the low PPV and, consequently, the fact that a significant fraction of seropositive cases detected in serosurveys may be false positives. Note, this conclusion does not depend on disease surveillance data, as our model does not incorporate these endpoints, and our study focuses exclusively on seroprevalence data. We acknowledge the limitations of disease surveillance data, which we discussed in the introduction. Also, we agree with the reviewer that seroprevalence data are useful (as they quantify exposure), and our study merely shows a caveat to their interpretation. As discussed above, reality must sit somewhere in the middle, so we propose that the way forward is to incorporate all available information within a single modeling framework. That said, we acknowledge that certain aspects of the text may appear

overly critical of seroprevalence data. In response, we revised several parts of the manuscript, including the abstract and the discussion (fifth paragraph).

LINES 470-475: similarly, while I agree with your conclusion that immune boosting is a factor to consider as limitation in quantifying transmissible infection from seroprevalence studies, I'm not sure it is valid to estimate PPV since ρ could not be estimated.

The unidentifiability of the immune boosting parameter is indeed an important limitation of our study, which we acknowledged in the discussion:

“Finally, even though we tested a range of realistic values, we could not estimate the strength of immune boosting, a parameter that sensitively controls the prevalence and reliability of seropositivity. As noted above, additional information in the form of case-based data should help resolve this uncertainty. Acknowledging these limitations, our model could serve as a building block to investigate the remaining unknowns in pertussis epidemiology.”

As discussed above, however, we considered a range (0.5–5) of values for this parameter, including a scenario of hyposensitive boosting ($\rho = 0.5$), which—if we correctly understand the reviewer’s comment on lines 166–167—may be immunologically unlikely and too unfavorable to our working hypothesis. We also note that some estimates of ρ were higher than the upper bound we considered (e.g., point estimate of 6.6 in [7] and estimate of ≥ 10 in [6]). Thus, we believe our conclusions regarding the PPV are justified, although we recognize that future research should aim to estimate the ρ parameter more precisely.

LINES 477-485: I can certainly agree that comparing case notification data and seroprevalence data is fraught with biases. However, for the same reasons, it is as inadequate to discount the evidence of serosurveys vs case surveillance. Case surveillance is well documented to under- estimate disease burden, and completely misrepresents the dynamics of transmissible infections. Both data are interesting in that they reflect, albeit imperfectly, different aspects of the pathogen dynamics: circulation vs. disease

As discussed above, our results do not imply that seroprevalence data should be dismissed; however, they highlight a caveat to their interpretation. We agree with the reviewer that case surveillance data have limitations; however, in some cases, these limitations can be overcome by using appropriate transmission models with age-specific case reporting rates (see, e.g., [1]). We agree that both data sources are interesting, and we propose that the way forward is to synthesize evidence from all available data within a single modeling framework. We have added this sentence to lines 477–478 to clarify this point:

“Therefore, our results do not imply that serosurveys are irrelevant, as seroprevalence data can accurately quantify the level of pertussis exposure in the population.”

LINES 494-495: While the skewing of the immune response is comparable between natural infection and wP vaccines, I'm not sure it is correct to say the immune response is similar as

one is led by mucosal stimulation whereas the other is parenteral. In humans, data is difficult to generate/interpret data on this topic. The baboons challenge data do suggest a difference in effect on colonization between convalescent and wP vaccinated baboons.

In response, we toned down this statement, using the adjective “comparable” instead of “similar”.

LINES 498-499: What about the interplay between a differential effect on transmission and observed duration of protection? Is it possible that difference in immune response between aP and wP vaccines may lead to a differential herd effect that could translate at population level into a different force of infection and thus an apparent difference in duration of protection, without there necessarily being one. Could your model help explore this?

This is a very interesting idea. In our model, two durations of protection can be defined and estimated: the “intrinsic” duration, in the absence of immune boosting ($\rho = 0$), and the “apparent” duration, in the presence of immune boosting. Due to exposures and boosting events, the apparent duration will always exceed the intrinsic duration. As aP and wP likely have different effects on the force of infection and herd immunity (which determine the frequency of exposures), the difference in the apparent duration for the two vaccines may indeed be biased and not reflect their intrinsic properties. Our model can be used to explore this issue; however, as we explain in the discussion, this will require extending the model to represent aP-derived immunity explicitly. Although beyond the scope of this study, we plan to do so in the future.

Reviewer #4 (Remarks to the Author): The article presents a model to assess the reliability of seropositivity as a marker for pertussis infection. It distinguishes between seropositivity, symptomatic and asymptomatic infections, and highlights immune boosting—exposure triggering a detectable response without transmission—as a source of overestimation in serosurveys. The authors address these ambiguities in the literature through a literature review. Then, the authors propose a model that integrates infection, immune boosting, and seropositivity dynamics, accounting for anti-pertussis toxin IgG titers, as well as demographic and social factors. Calibrated with late whole-cell vaccine era data from six European countries, the model reveals a weak correlation between infection and seropositivity, reflected in a low positive predictive value (PPV), particularly among younger populations likely due to routine vaccine boosters and higher contact rates that favor transmission. Waning immunity and social contact patterns also impact reliability, underscoring the need for refined tools to interpret seroprevalence data. The article is well written and in general clear.

Several key questions emerge:

1. Regarding the literature review, the authors should provide a clearer description of the methodology used, including the search strategy, whether study selection was performed by one or multiple reviewers, data extraction, and quality assessment. Additionally, they should clarify how they adhered to the PRISMA-P checklist, and I recommend including the checklist as supplementary material.

The reviewer raises an important point. The methodology of the literature review was previously described in the supplementary materials. In response to Reviewer 1, we have now relocated this paragraph to the method section in the main text. The study selection was performed by one reviewer (Anabelle Wong), while the data extraction was performed by two reviewers (Anabelle Wong and Matthieu Domenech). Note, the quality assessment was not required for this review, as we assessed a textual and non-numerical outcome: how investigators interpreted their seroprevalence data. Finally, we now provide the PRISMA-P checklist as a supplement.

2. More detailed explanations are needed regarding the definitions used—such as exposure, immune boosting, and seropositivity—and the literature supporting these distinctions should be cited.

As explained in Methods/Model structure, we used the following definition of seropositivity:

“Unless otherwise stated, seropositivity was defined as an antibody titer exceeding 100 IU/mL, a standard threshold used in numerous seroprevalence studies (Refs. [23,47] and Table S2).”

In the introduction, we gave the following definitions for exposure, infection, and immune boosting:

“To clarify our terminology regarding the outcomes of *B. pertussis* exposures resulting in seropositivity, we henceforth restrict our definition of infection to an exposure leading to transmissible infection (either symptomatic or asymptomatic), and we define other exposures as subclinical immune boosts. We believe this definition is justified, as transmissible infections are arguably the most pertinent from an epidemiological and evolutionary perspective.”

Admittedly, these definitions are *ad hoc* (though unambiguous in the context of our model). However, to the best of our knowledge, no official definition of these terms exists in the literature. Arguably, the lack of a clear definition of infection is the primary cause of confusion regarding the interpretation of serosurveys.

3. The model was calibrated using a survey from the late wP vaccine era rather than a more recent one, which raises several questions:

- How realistic is the assumption that vaccine-derived immunity, particularly from whole-cell vaccines, is imperfect and may result in immediate failure (primary vaccine failure), followed by waning immunity mentioned in the Introduction?

This assumption was based on extensive fits of transmission models to pertussis incidence data in Massachusetts, USA [1]. In this study, we systematically tested a range of assumptions regarding the mode and degree of pertussis vaccine failure. The best model included a combination of primary vaccine failure and subsequent waning immunity. Hence, this assumption was based on empirical data, and we note that other modeling studies made similar assumptions [15]. More broadly, we argue that this assumption is consistent with robust evidence from the field of pertussis vaccinology showing that wP vaccine effectiveness is less than 100% immediately after vaccination and subsequently declines over time since vaccination.

- How might the findings differ in the acellular pertussis (aP) vaccine era, and how can the model be applied in countries using the aP vaccines (most countries now)?

This is a very important question that we could not address in this study, but it would be an important line of future research. As explained in the discussion, analysis of data during the aP era would require extending our model to represent aP-derived immunity:

“Hence, extending our model to the aP era would require a separate parametrization for infection/wP-derived aP-derived immunities. This complication prevented us from considering more recent seroprevalence data, especially those from another large serosurvey by Berbers et al. in adults aged 40–59 conducted in 18 European countries in 2015 [43].”

4. The cyclical pattern as well as seasonality of pertussis may influence seroprevalence—how was this accounted for in the model? Additionally, were differences in vaccine booster policies across the studied countries considered?

Our model did not account for seasonality in transmission, which is indeed an important limitation that we acknowledged in the discussion. However, in all countries included in our analysis, sample collection took place over ~1 year, so seasonal effects would have been smoothed out in the data. We revised the text to clarify this point:

“Second, as discussed above, when fitting our model to seroprevalence data, we did not consider seasonal or long-term parameter variations that may affect seroprevalence. However, in all countries included in our analysis, sample collection took place over ~1 year, allowing for any seasonal effects, if present, to be smoothed out.”

Regarding the cyclical pattern, our model reproduced it in all scenarios (see Fig. S5). However, as we didn't include other data about past epidemics, our model could not capture the time of the epidemic cycle at which the serosurvey was conducted. Hence, our model predictions were based on averages across multiple cycles. We note that, for the countries included in our analysis, the seroprevalence was relatively consistent, ranging from 1% to 3%. This could suggest that the serosurvey was conducted at a similar stage of the epidemic cycle in each country. Again, as we emphasized in the manuscript, the model-data comparison was only approximate, but our estimates are consistent with earlier evidence. We revised the text (Discussion) to summarize these points:

“In addition, we did not include other data to inform about the past epidemics and age-specific history of prior exposures and infections in each country. In practice, the serosurveys may have captured a different stage of the epidemic cycle in every country, yet our model predictions were based on averages across multiple epidemic cycles (Fig. S5).”

Regarding the differences in booster policies, they were indeed considered for the model-data comparison (see Table S6).

5. Can the model integrate other factors such as social changes in the population, non-pharmaceutical mitigation measures, macrolide resistance etc.?

Though beyond the scope of this study, it is indeed possible to consider other factors in our model. For example, social changes could be modeled by varying social contact patterns or demography over time. Similarly, non-pharmaceutical measures could be modeled by varying the transmission rate during the early stages of the pandemic. Regarding macrolide resistance, considering this aspect would likely require a substantial change to our model. However, models integrating resistance are relatively standard for other pathogens, such as *S. pneumoniae*, and these models could also be applied to pertussis.

6. The authors should specify how the model could be used, for which populations it would be most reliable, and how it could be integrated with other sources to identify transmission.

The model itself is generic and can be applied to any population. In practice, however, parametrization will be population-specific, as we expect multiple model parameters (such as social contact patterns, transmission levels, and demographic structure) to vary across countries. In our analysis, we considered a total of sixteen countries by varying some of these parameters (see Figs. S3–S4). The model's reliability will depend on the quality of the data used for parametrization. Hence, to the extent that good data are available, we envision our model could be used in multiple populations to synthesize evidence from multiple data streams and estimate transmission and infection levels. We develop this idea in the conclusion of our paper:

“Hence, our model—or variations thereof—may prove useful for analyzing seroprevalence data and synthesizing evidence from other sources, including case notification data. Eventually, fitting such models to all available data in multiple populations will improve our estimates of pertussis infection rates and help resolve the ongoing disagreements within the field.”

References

1. Domenech de Cellès M, Magpantay FMG, King AA, Rohani P. The impact of past vaccination coverage and immunity on pertussis resurgence. *Sci Transl Med*. 2018;10. doi:10.1126/scitranslmed.aaj1748
2. Luo Y, Xiao X, Jiang Y, Li X, Ge L, Hu Y, et al. Global, regional, and national burden of pertussis in children aged 0–14 years from 1990–2021: A systematic analysis for the global burden of disease study 2021. *Social Science Research Network*. 2024. doi:10.2139/ssrn.4986351
3. Shaw D, Abad R, Amin-Chowdhury Z, Bautista A, Bennett D, Broughton K, et al. Trends in invasive bacterial diseases during the first 2 years of the COVID-19 pandemic: analyses of prospective surveillance data from 30 countries and territories in the IRIS Consortium. *Lancet Digit Health*. 2023;5: e582–e593. doi:10.1016/S2589-7500(23)00108-5
4. Ryan M, Murphy G, Ryan E, Nilsson L, Shackley F, Gothefors L, et al. Distinct T-cell subtypes induced with whole cell and acellular pertussis vaccines in children. *Immunology*. 1998;93: 1–10. doi:10.1046/j.1365-2567.1998.00401.x
5. Eberhardt CS, Siegrist C-A. What Is Wrong with Pertussis Vaccine Immunity? Inducing and Recalling Vaccine-Specific Immunity. *Cold Spring Harb Perspect Biol*. 2017;9. doi:10.1101/cshperspect.a029629
6. Lavine JS, King AA, Bjørnstad ON. Natural immune boosting in pertussis dynamics and the potential for long-term vaccine failure. *Proc Natl Acad Sci U S A*. 2011;108: 7259–7264. doi:10.1073/pnas.1014394108
7. Lavine JS, King AA, Andreasen V, Bjørnstad ON. Immune boosting explains regime-shifts in prevaccine-era pertussis dynamics. *PLoS One*. 2013;8: e72086. doi:10.1371/journal.pone.0072086
8. de Greeff SC, de Melker HE, van Gageldonk PGM, Schellekens JFP, van der Klis FRM, Mollema L, et al. Seroprevalence of pertussis in The Netherlands: evidence for increased circulation of *Bordetella pertussis*. *PLoS One*. 2010;5: e14183. doi:10.1371/journal.pone.0014183
9. Campbell P, McIntyre P, Quinn H, Hueston L, Gilbert GL, McVernon J. Increased population prevalence of low pertussis toxin antibody levels in young children preceding a record pertussis epidemic in Australia. *PLoS One*. 2012;7: e35874. doi:10.1371/journal.pone.0035874
10. Wearing HJ, Rohani P. Estimating the duration of pertussis immunity using epidemiological signatures. *PLoS Pathog*. 2009;5: e1000647. doi:10.1371/journal.ppat.1000647
11. Lee AD, Cassidy PK, Pawloski LC, Tatti KM, Martin MD, Briere EC, et al. Clinical evaluation and validation of laboratory methods for the diagnosis of *Bordetella pertussis* infection: Culture, polymerase chain reaction (PCR) and anti-pertussis toxin IgG serology (IgG-PT). *PLoS One*. 2018;13: e0195979. doi:10.1371/journal.pone.0195979

12. Storsaeter J, Hallander HO, Gustafsson L, Olin P. Low levels of antipertussis antibodies plus lack of history of pertussis correlate with susceptibility after household exposure to *Bordetella pertussis*. *Vaccine*. 2003;21: 3542–3549. doi:10.1016/s0264-410x(03)00407-9
13. Graaf H de, de Graaf H, Ibrahim M, Hill AR, Gbesemete D, Vaughan AT, et al. Controlled Human Infection With *Bordetella pertussis* Induces Asymptomatic, Immunizing Colonization. *Clin Infect Dis*. 2020;71: 403–411. doi:10.1093/cid/ciz840
14. Teunis PFM, van Eijkeren JCH, de Graaf WF, Marinović AB, Kretzschmar MEE. Linking the seroresponse to infection to within-host heterogeneity in antibody production. *Epidemics*. 2016;16: 33–39. doi:10.1016/j.epidem.2016.04.001
15. Gambhir M, Clark TA, Cauchemez S, Tartof SY, Swerdlow DL, Ferguson NM. A change in vaccine efficacy and duration of protection explains recent rises in pertussis incidence in the United States. *PLoS Comput Biol*. 2015;11: e1004138. doi:10.1371/journal.pcbi.1004138